# Meek Separators
# and Their Applications in Targeted Causal Discovery

**Kirankumar Shiragur**[*]
Eric and Wendy Schmidt Center,
Broad Institute of MIT and Harvard

**Jiaqi Zhang**[*]
LIDS, Massachusetts Institute of Technology
Broad Institute of MIT and Harvard

**Caroline Uhler**
LIDS, Massachusetts Institute of Technology
Broad Institute of MIT and Harvard

## Abstract

Learning causal structures from interventional data is a fundamental problem with broad applications across various fields. While many previous works have focused on recovering the entire causal graph, in practice, there are scenarios where learning only part of the causal graph suffices. This is called *targeted* causal discovery. In our work, we focus on two such well-motivated problems: subset search and causal matching. We aim to minimize the number of interventions in both cases.

Towards this, we introduce the *Meek separator*, which is a subset of vertices that, when intervened, decomposes the remaining unoriented edges into smaller connected components. We then present an efficient algorithm to find Meek separators that are of small sizes. Such a procedure is helpful in designing various divide-and-conquer-based approaches. In particular, we propose two randomized algorithms that achieve logarithmic approximation for subset search and causal matching, respectively. Our results provide the first known average-case provable guarantees for both problems. We believe that this opens up possibilities to design near-optimal methods for many other targeted causal structure learning problems arising from various applications.

## 1 Introduction

Discovering the causal structure among a set of variables is an important problem permeating multiple fields including biology, epidemiology, economics, and social science [FLNP00, RHB00, SGSH00, Pea03]. A common way to represent the causal structure is through a *directed acyclic graph* (DAG), where an arc between two variables encodes a direct causal effect [SGSH00]. The goal of causal discovery is thus to recover this DAG from data. With observational data, a DAG is generally only identifiable up to its *Markov equivalence class* (MEC) [VP90, AMP97]. Identifiability can be improved by performing interventions on the variables. In particular, a more refined MEC can be identified with both hard and soft interventions [HB12, YKU18], where a *hard* intervention eliminates the dependency between its target variables and their parents in the DAG and a *soft* intervention modifies this dependency without removing it [ES07].

As intervention experiments tend to be expensive in practice, a critical problem is to design algorithms to select interventions that minimize the number of trials needed to learn about the structure. Previous works have considered both fully identifying the DAG while minimizing the total number/cost of interventions [HB14, SKDV15, KDV17, GKS+19, SMG+20] and learning the most about the

---

[*]Equal contributions. Alphabetical order.

37th Conference on Neural Information Processing Systems (NeurIPS 2023).

underlying DAG given a fixed budget [GSKB18]. While recovering the entire causal graph yields a holistic view of the relationships between variables, it is sometimes sufficient to learn only part of the causal graph for a particular downstream task. This is sometimes termed *targeted causal discovery* [SU22] and it has arisen in various different applications recently [ASY$^+$19, GHD20, ZSU21, CS23]. The benefit of targeted causal discovery is that the number of required interventions can be significantly less than that needed to fully identify the DAG. In our work, we consider two such problems, *subset search* and *causal matching*, described in the following.

**Subset Search.** Proposed by [CS23], the problem of subset search aims to recover a subset of the causal relationships between variables. Formally, let $\mathcal{G} = (V, E)$ denote the underlying DAG. Given a subset of target edges $T \subseteq E$, the goal is to recover the orientation of edges in $T$ with the minimum number of interventions. Subset search problems arise in many applications, including local graph discovery for feature selection [ATS03], out-of-distribution generalization in machine learning [HDPM18], and learning gene regulatory networks [DL05]. As a concrete example, consider the study of melanoma [FMT$^+$21], a type of skin cancer. To understand its development for potential treatments, it may be of interest to identify the causal relationships between genes that are known to be relevant to melanoma. In this case, the problem can be formulated as recovering the subset of edges between melanoma-related genes.

**Causal Matching.** Motivated by many sequential design problems, [ZSU21] considered causal matching, where the goal is to identify an intervention which transforms a causal system to a desired state. Formally, let $V$ be the system variables and $P$ be its joint distribution which factorizes according to the underlying DAG $\mathcal{G}$. Given a desired joint distribution $Q$ over $V$, the goal is to identify an intervention $I$ such that the interventional distribution $P^I$ best matches $Q$ under some metric. Similar to [ZSU21], we will focus on a special form of this problem, *causal mean matching*, where the metric between $P^I$ and $Q$ is characterized by the mean discrepancy $\|\mathbb{E}_{P^I}[V] - \mathbb{E}_Q[V]\|_2$. Such problems arise in, for example, cellular reprogramming in genomics, which is of great interest for regenerative medicine [CD12]. The aim of this field is to reprogram easily accessible cell types into some desired cell types using genetic interventions. Since genes regulate each other via an underlying network, a targeted strategy can infer just enough about the structure in order to match the desired state.

**Contributions.** One of our primary contributions is an efficient algorithm for finding an intervention set $\mathcal{I}$ of small size that, when intervened, decomposes the remaining unoriented edges into connected components of smaller sizes. This procedure is powerful in its flexibility to be used to design various divide-and-conquer based algorithms. In particular, we demonstrate this on the two targeted causal discovery problems of subset search and causal matching.[2]

- For the subset search problem, we obtain an efficient randomized algorithm that in expectation achieves a logarithmic approximation to the minimum number of required interventions (verification number; defined in Section 2).
- For the causal mean matching problem, we derive a randomized algorithm that on average achieves a logarithmic competitive ratio against the optimal number of required interventions.

For both problems, we obtain *exponential* improvements – our analysis gives the first non-trivial competitive ratio in expectation; in contrast, prior works [CS23, ZSU21] show that no deterministic algorithm can achieve a better than linear approximation against the optimal solution for all instances.

Philosophically, the results in our work are a step towards the study of targeted causal discovery problems arising from various applications, where recovering the entire causal graph is unnecessary.

**Organization.** In Section 2, we provide formal definitions and useful results. In Section 3, we state all our main results. We provide our algorithm for Meek separator and its analysis in Section 4. We use the Meek separator subroutine to solve for subset search in Section 5 and causal matching in Section 6. In Section 7, we demonstrate empirically our proposed algorithms on synthetic data.

## 2 Preliminaries and Related Work

### 2.1 Basic Graph Definitions

Let $\mathcal{G} = (V, E)$ be a graph on $|V| = n$ vertices. We use $V(\mathcal{G})$, $E(\mathcal{G})$ and $A(\mathcal{G}) \subseteq E(\mathcal{G})$ to denote its vertices, edges, and oriented arcs respectively. When the referred graph is clear from the

---

[2]In this work, we assume causal sufficiency [SGSH00] and noiseless setting with given observational data.

context, we use $V$ (or $E$) instead of $V(\mathcal{G})$ (or $E(\mathcal{G})$) for simplicity. The graph $\mathcal{G}$ is fully oriented if $A(\mathcal{G}) = E(\mathcal{G})$, and partially oriented otherwise. For any two vertices $u, v \in V$, we write $u \sim v$ if they are adjacent and $u \not\sim v$ otherwise. To specify the arc orientations, we use $u \to v$ or $u \leftarrow v$. For any subset $V' \subseteq V$ and $E' \subseteq E$, let $\mathcal{G}[V']$ and $\mathcal{G}[E']$ denote the vertex-induced and edge-induced subgraphs respectively. Consider a vertex $v \in V$ in a directed graph, let $\mathtt{Pa}(v)$, $\mathtt{Anc}(v)$ and $\mathtt{Des}(v)$ denote the parents, ancestors and descendants of $v$ respectively. Let $\mathtt{Des}[v] = \mathtt{Des}(v) \cup \{v\}$ and $\mathtt{Anc}[v] = \mathtt{Anc}(v) \cup \{v\}$. The *skeleton* $skel(\mathcal{G})$ of a graph $\mathcal{G}$ refers to the graph where all edges are made undirected. A *v-structure* refers to three distinct vertices $u, v, w$ such that $u \to v \leftarrow w$ and $u \not\sim w$. A *simple cycle* is a sequence of $k \geq 3$ vertices where $v_1 \sim v_2 \sim \ldots \sim v_k \sim v_1$. The cycle is directed if at least one of the edges is directed and all oriented arcs are in the same direction along the cycle. A partially oriented graph is a *chain graph* if it contains no directed cycle. In the undirected graph $\mathcal{G}[E \setminus A]$ obtained by removing all arcs from a chain graph $\mathcal{G}$, each connected component in $\mathcal{G}[E \setminus A]$ is called a *chain component*. We use $CC(\mathcal{G})$ to denote the set of all such chain components, where each chain component $\mathcal{H} \in CC(\mathcal{G})$ is a subgraph of $\mathcal{G}$ and $V = \dot{\cup}_{\mathcal{H} \in CC(\mathcal{G})} V(\mathcal{H})$.[3] For any partially oriented graph, an *acyclic completion* or *consistent extension* refers to an assignment of orientations to undirected edges such that the resulting fully oriented graph has no directed cycles.

## 2.2 Graphical Concepts in Causal Models

Directed acyclic graphs (DAGs) are fully oriented chain graphs, where vertices represent random variables and their joint distribution $P$ factorizes according to the DAG: $P(v_1, \ldots, v_n) = \prod_{i=1}^{n} P(v_i \mid \mathtt{Pa}(v))$. We can associate a *valid permutation* or *topological ordering* $\pi : V \to [n]$ to any (partially oriented) DAG such that oriented arcs $u \to v$ satisfy $\pi(u) < \pi(v)$ (and assigning undirected edges $u \sim v$ as $u \to v$ when $\pi(u) < \pi(v)$ is an acyclic completion). Note that such valid permutation is not necessarily unique. Two DAGs $\mathcal{G}_1, \mathcal{G}_2$ are in the same *Markov equivalence class* (MEC) if any positive distribution $P$ which factorizes according to $\mathcal{G}_1$ also factorizes according $\mathcal{G}_2$. For any DAG $\mathcal{G}$, we denote its MEC by $[\mathcal{G}]$. It is known that DAGs in the same MEC share the same skeleton and v-structures [VP90, AMP97]. A *moral* DAG is a DAG without v-structures. Figure 1 illustrates this definition. The *essential graph* $\mathcal{E}(\mathcal{G})$ of $[\mathcal{G}]$ is a partially oriented graph such that an arc $u \to v$ is oriented if $u \to v$

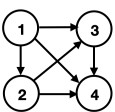

(a) A moral DAG.

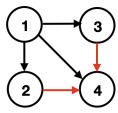

(b) A DAG with v-structure $2 \to 4 \leftarrow 3$.

Figure 1: An example of moral DAGs. **(a)** a moral DAG without v-structures. **(b)** an *im*moral DAG.

in *every* DAG in MEC $[\mathcal{G}]$, and an edge $u \sim v$ is undirected if there exists two DAGs $\mathcal{G}_1, \mathcal{G}_2 \in [\mathcal{G}]$ such that $u \to v$ in $\mathcal{G}_1$ and $u \leftarrow v$ in $\mathcal{G}_2$. An edge $u \sim v$ is a *covered edge* [Chi95, Definition 2] if $\mathtt{Pa}(u) \setminus \{v\} = \mathtt{Pa}(v) \setminus \{u\}$. We now give some useful definition and result for graph separators:

**Definition 1** ($\alpha$-separator and $\alpha$-clique separator [CSB22]). Let $A, B, C$ be a partition of the vertices $V$ of a graph $\mathcal{G}$. We say that $C$ is an *$\alpha$-separator* if no edge joins[4] a vertex in $A$ with a vertex in $B$ and $|A|, |B| \leq \alpha \cdot |V|$. We call $C$ is an *$\alpha$-clique separator* if it is an *$\alpha$-separator* and a clique.[5]

**Lemma 2** (Theorem 1,3 in [GRE84], instantiated for unweighted graphs). *Let $\mathcal{G} = (V, E)$ be a chordal graph[6] with $|V| \geq 2$. Denote $\omega(\mathcal{G})$ as the size of its largest clique. There exists a $\mathsf{1}/\mathsf{2}$-clique separator $C$ involving at most $\omega(\mathcal{G}) - 1$ vertices. The clique $C$ can be computed in $\mathcal{O}(|E|)$ time.*

## 2.3 Interventions and Verifying Sets

An *intervention* $I \subseteq V$, associated with random variables $V$ with joint distribution $P$ that factorizes according to $\mathcal{G}$, is an experiment where the conditional distributions $P(v \mid \mathtt{Pa}(v))$ for $v \in I$ are changed. *Hard* interventions refer to changes that eliminate the dependency between $v$ and $\mathtt{Pa}(v)$, while *soft* interventions modify this dependency without removing it [ES07]. Let $P^I$ denote the interventional distribution. Observational data is a special case where $I = \varnothing$. An intervention is *atomic* if $|I| = 1$ and *bounded* if $|I| \leq k$. We call a set of interventions $\mathcal{I} \subseteq 2^V$ an *intervention set*.

---

[3]We denote $A \dot{\cup} B$ as the union of two disjoint sets $A$ and $B$.

[4]Edge $u \sim v$ *joins* vertex $u$ with vertex $v$.

[5]A clique is a fully connected graph, i.e., $u \sim v$ for every pair of distinct vertices in the graph.

[6]A chordal graph is a graph where every cycle of length at least 4 has a chord (c.f. [BP93]).

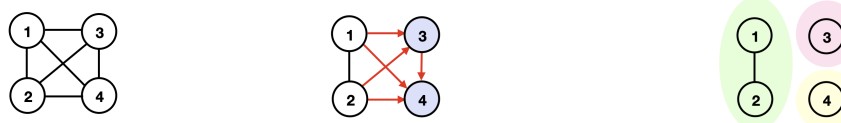

(a) Essential graph $\mathcal{E}(\mathcal{G})$.     (b) $\mathcal{E}_{\mathcal{I}}(\mathcal{G})$ with $\mathcal{I} = \{\{3\}, \{4\}\}$.     (c) Connected components of $CC(\mathcal{E}_{\mathcal{I}}(\mathcal{G}))$.

Figure 2: An example illustrating essential graphs and connected chain components. Suppose the ground-truth DAG is given in Fig. 1a. **(a)** the essential graph of $\mathcal{G}$. **(b)** the $\mathcal{I}$-essential graph of $\mathcal{G}$ after two atomic interventions $\mathcal{I} = \{\{3\}, \{4\}\}$, where edges oriented by $\mathcal{I}$ are indicated in red. **(c)** three connected components of $\mathcal{E}_{\mathcal{I}}(\mathcal{G})$ after removing the oriented edges.

With observational data, a DAG $\mathcal{G}$ is *generally* only identifiable up to its MEC[7], i.e., $\mathcal{E}(\mathcal{G})$ [AMP97]. Identifiability can be improved with interventional data and it is known that intervening on $I$ allows us to infer the edge orientation of any edge cut by $I$ and $V \setminus I$ and possibly additional edges given by the Meek rules (Appendix A), for both hard [HB12] and soft [YKU18] interventions.[8] For intervention set $\mathcal{I}$, let the *$\mathcal{I}$-essential graph* $\mathcal{E}_{\mathcal{I}}(\mathcal{G})$ of $\mathcal{G}$ be the essential graph representing all DAGs in $[\mathcal{G}]$ whose orientations of arcs cut by $I$ and $V \setminus I$ are the same as $\mathcal{G}$ for all $I \in \mathcal{I}$. Figure 2 illustrates these concepts. The aforementioned results state that $\mathcal{G}$ can be identified up to $\mathcal{E}_{\mathcal{I}}(\mathcal{G})$ with observational and interventional data from $\mathcal{I}$. We state some useful properties about $\mathcal{I}$-essential graphs from [HB14]. First, every $\mathcal{I}$-essential graph is a chain graph with chordal chain components. This includes the case of $\mathcal{I} = \varnothing$. Second, orientations in one chain component do not affect orientations in other components. In other words, to fully orient any essential graph $\mathcal{E}_{\mathcal{I}}(\mathcal{G})$, it is necessary and sufficient to orient every chain component in $\mathcal{E}_{\mathcal{I}}(\mathcal{G})$.

A *verifying set* $\mathcal{I}$ for a DAG $\mathcal{G}$ [CSB22] is an intervention set that fully orients $\mathcal{G}$ from $\mathcal{E}(\mathcal{G})$, possibly with repeated applications of Meek rules (see Appendix A). In other words, for any graph $\mathcal{G} = (V, E)$ and any verifying set $\mathcal{I}$ of $\mathcal{G}$, we have $\mathcal{E}_{\mathcal{I}}(\mathcal{G}) = \mathcal{G}$. A *subset verifying set* $\mathcal{I}$ [CS23] for a subset of *target edges* $T \subseteq E$ in a DAG $\mathcal{G}$ is an intervention set that fully orients all arcs in $T$ given $\mathcal{E}(\mathcal{G})$, possibly with repeated applications of Meek rules. Note that the subset verifying set depends on the target edges *and* the underlying ground truth DAG — the subset verifying set for the same $T \subseteq E$ may differ across two different DAGs $\mathcal{G}_1, \mathcal{G}_2$ in the same Markov equivalence class.

For bounded interventions of size at most $k$, the *minimum verification number* $\nu_k(\mathcal{G}, T)$ denotes the size of the minimum size subset verifying set for any DAG $\mathcal{G} = (V, E)$ and subset of target edges $T \subseteq E$. We write $\nu_1(\mathcal{G}, T)$ when we restrict to atomic interventions. When $k = 1$ and $T = E$ (i.e., full graph identification), [CSB22] showed that it is necessary and sufficient to intervene on a minimum vertex cover of the covered edges in $\mathcal{G}$. For any intervention set $\mathcal{I} \subseteq 2^V$, we denote $R(\mathcal{G}, \mathcal{I}) = A(\mathcal{E}_{\mathcal{I}}(\mathcal{G})) \subseteq E$ as the set of oriented arcs in the $\mathcal{I}$-essential graph of a DAG $\mathcal{G}$. For cleaner notation, we write $R(\mathcal{G}, I)$ for one intervention $\mathcal{I} = \{I\}$ for some $I \subseteq V$, and $R(\mathcal{G}, v)$ for one atomic intervention $\mathcal{I} = \{\{v\}\}$ for some $v \in V$.

**Definition 3.** For any intervention set $\mathcal{I} \subseteq 2^V$, define $\mathcal{G}^{\mathcal{I}} = \mathcal{G}[E \setminus R(G, \mathcal{I})]$ as the *fully oriented* subgraph of $\mathcal{G}$ induced by the unoriented edges in $\mathcal{E}_{\mathcal{I}}(\mathcal{G})$. In addition, for $u \in V$, let $\mathrm{Pa}_{\mathcal{G}, \mathcal{I}}(u) = \{x \in V : x \to u \in R(G, \mathcal{I})\}$ as the *recovered parents* of $u$ by $\mathcal{I}$.

## 3 Results

Here we state all the main results of the paper. One of the primary contributions of our work is a randomized algorithm that outputs an intervention set $\mathcal{I}$ of small size such that all connected components in the resulting $\mathcal{I}$-essential graph have small sizes. We now formally define such intervention sets as Meek separators. An example of $1/2$-Meek separator is shown in Figure 3.

**Definition 4** ($\alpha$-Meek separator). We call an intervention set $\mathcal{I}$ an *$\alpha$-Meek separator* of $\mathcal{G}$ if each connected component $\mathcal{H} \in CC(\mathcal{E}_{\mathcal{I}}(\mathcal{G}))$ satisfies: $|V(\mathcal{H})| \leq \alpha |V(\mathcal{G})|$.

Note that Meek separators differ from the traditional graph separators in Definition 1, where in the latter we have bounds on the sizes of connected components in $\mathcal{G} \setminus \mathcal{I}$ instead of $CC(\mathcal{E}_{\mathcal{I}}(\mathcal{G}))$. Graph separators of small size may not exist. For example, consider a fully connected DAG $\mathcal{G} = (V, E)$

---

[7]With additional parametric assumptions (e.g., [PB14]), additional identifiability can be achieved.

[8]For both cases, a "faithfulness" assumption needs to be assumed for $P, P^I$ (c.f. [YKU18]).

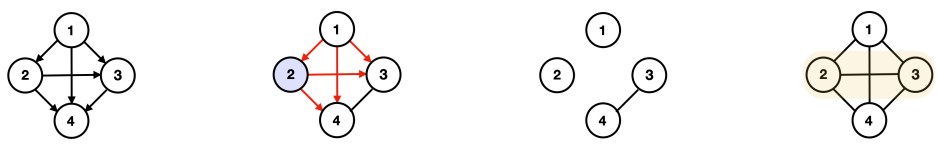

(a) DAG $\mathcal{G}$.    (b) $1/2$-Meek separator $\mathcal{I} = \{\{2\}\}$.    (c) $CC(\mathcal{E}_{\mathcal{I}}(\mathcal{G}))$.    (d) $1/2$-graph separator $\{2, 3\}$.

Figure 3: An example comparing a $1/2$-Meek separator with a $1/2$-graph separator. **(a)** the underlying true DAG $\mathcal{G}$. **(b)** the essential graph $\mathcal{E}_{\mathcal{I}}(\mathcal{G})$ after intervening on the $1/2$-Meek separator $\mathcal{I} = \{\{2\}\}$, where edges oriented are indicated in red. **(c)** connected components of size $\leq |V(\mathcal{G})|/2$ in $\mathcal{E}_{\mathcal{I}}(\mathcal{G})$. **(d)** the undirected version of $\mathcal{G}$ with the $1/2$-graph separator $\{2, 3\}$ highlighted.

that forms a clique. Any $\alpha$-graph separator in $\mathcal{G}$ must contain at least $(1 - \alpha)|V|$ vertices, since every pair of vertices in the clique is connected. In contrast, we show that small-sized Meek separators always exist for any DAG. Moreover, we can efficiently find such separators by performing very few interventions, as given by the following theorem which we prove in Section 4.

**Theorem 5** (Meek separator). *Given an essential graph $\mathcal{E}(\mathcal{G})$ of an unknown DAG $\mathcal{G}$, there exists a randomized procedure* MeekSeparator *(given in Algorithm 1) that runs in polynomial time and adaptively intervenes on a set of atomic interventions $\mathcal{I}$ such that we can find a $1/2$-Meek separator of size at most 2 and $\mathbb{E}\left[|\mathcal{I}|\right] \leq \mathcal{O}(\log \omega(\mathcal{G}))$,[9] where $\omega(\mathcal{G})$ denotes the size of the largest clique in $\mathcal{G}$.*

Although in the above result, we intervene on $\mathcal{O}(\log \omega(\mathcal{G}))$ nodes, our proofs show that there always exists a Meek separator of size at most 2. However, to find such a Meek separator without knowing $\mathcal{G}$ *a priori*, our algorithm needs to perform $\mathcal{O}(\log \omega(\mathcal{G}))$ many interventions. The above result is significant because it can be used to design divide-and-conquer based approaches for various problems. Specifically, we use the Meek separator algorithm as a subroutine to develop approximation algorithms for the subset search and causal matching problems.

**Subset Search.** Our first application of the Meek separator result is that it can be used as a subroutine for approximately solving the subset search problem, as demonstrated in Algorithm 2. The analysis of our algorithm for subset search consists of two parts: (1) a lower bound on the number of interventions required for any algorithm (even with knowledge of $\mathcal{G}$) to solve subset search, and (2) an upper bound on the number of interventions needed for our algorithm to solve subset search. By combining the two parts, we can bound the competitive ratio of our algorithm.

For the first part, we provide a lower bound for the subset verification number described below. Recall that the subset verification number is the minimum number of interventions needed to orient edges in $T$ by any algorithm with full knowledge of $\mathcal{G}$. Therefore it is a natural lower bound on the number of interventions required for any algorithm to solve subset search.

**Lemma 6** (Lower bound). *Let $\mathcal{G} = (V, E)$ be a DAG and $T \subseteq E$ be a subset of target edges, then,*
$$\nu_1(\mathcal{G}, T) \geq \max_{I \subseteq V} \sum_{\mathcal{H} \in CC(\mathcal{E}_I(\mathcal{G}))} \mathbb{1}(E(\mathcal{H}) \cap T \neq \varnothing) \, .$$

For the upper bound, we show that our randomized algorithm, designed using the Meek separator subroutine, is competitive with respect to the aforementioned lower bound. Thus, it achieves a logarithmic approximation to the optimal number of interventions required to solve subset search.

**Theorem 7** (Upper bound). *Let $\mathcal{G} = (V, E)$ be a DAG with $|V| = n$ and $T \subseteq E$ be a subset of target edges. Algorithm 2 that takes essential graph $\mathcal{E}(\mathcal{G})$ and $T$ as input, runs in polynomial time and adaptively intervenes on a set of atomic interventions $\mathcal{I} \subseteq V$ that satisfies, $\mathbb{E}\left[|\mathcal{I}|\right] \leq \mathcal{O}(\log n \cdot \log \omega(\mathcal{G})) \cdot \nu_1(\mathcal{G}, T)$ and $T \subseteq R(\mathcal{G}, \mathcal{I})$. Furthermore, in the special case of $T = E$, the solution returned by it satisfies, $\mathbb{E}\left[|\mathcal{I}|\right] \leq \mathcal{O}(\log n) \cdot \nu_1(\mathcal{G})$ and $E \subseteq R(\mathcal{G}, \mathcal{I})$.*

Importantly, our result provides the *first* known competitive ratio with respect to the subset verification number $\nu_1(\mathcal{G}, T)$. Prior to our work, a full-graph identification algorithm that is competitive with respect to $\nu_1(\mathcal{G}) = \nu_1(\mathcal{G}, E)$ was provided by [CSB22]. $\nu_1(\mathcal{G})$ is not a valid lower bound for the subset search problem. In particular, the value of $\nu_1(\mathcal{G}, T)$ for certain $T$ could be substantially smaller than $\nu_1(\mathcal{G})$. For instance, if $\mathcal{G}$ is a clique and $T$ is an edge incident to a particular node, then $\nu_1(\mathcal{G}, T) = 1$ while $\nu_1(\mathcal{G}) = n/2$. Hence, the benchmark based on the full-graph verification number can be significantly weaker compared to that based on the subset search verification number.

---

[9] The expectation in the result is over the randomness of the algorithm.

Additionally, for the subset search problem, [CS23] showed that there does *not* exist any *deterministic* algorithm that achieves a competitive ratio better than $\mathcal{O}(n)$ with respect to $\nu_1(\mathcal{G}, T)$. Therefore, our result shows that subset search is part of the large class of problems in algorithm design where *randomization* substantially helps. Finally, when $T = E$, we get a $\mathcal{O}(\log n)$ approximation for recovering the entire DAG, which matches the current best approximation ratio by [CSB22].

**Causal Matching.** Another application of our Meek separator result is for solving the causal matching problem. We consider the same setting as [ZSU21] (details provided in Section 6). In [ZSU21], it was shown that causal mean matching has a unique solution and can be solved by iteratively finding source vertices[10] of induced subgraphs of the underlying DAG (Lemma 1 and Observation 1 in [ZSU21]). We therefore provide an algorithm to find source vertices of any DAG, which can be used in the iterative process. Our algorithm uses the Meek separator result and is given in Algorithm 3; the iterative process of using this algorithm to solve causal mean matching is provided in Section 6 and Algorithm 4 in Appendix F.

Our analysis for causal mean matching consists of two layers. We first establish an upper bound on the number of interventions required in Algorithm 3 to identify a source vertex. Then we use this result within the iterative process to derive an upper bound on the number of interventions needed in Algorithm 4 to solve causal mean matching.

**Lemma 8** (Source finding). *Let $\mathcal{G} = (V, E)$ be a DAG and $U \subseteq V$ be a subset of vertices. Algorithm 3 that takes essential graph $\mathcal{E}(\mathcal{G})$ and $U$ as input, runs in polynomial time and adaptively intervenes on a set of atomic interventions $\mathcal{I} \subset V$, identifies a source vertex of the induced subgraph $\mathcal{G}[U]$ with $\mathbb{E}\left[|\mathcal{I}|\right] \leq \mathcal{O}\left(\log n \cdot \log \omega(\mathcal{G})\right)$.*

**Theorem 9** (Causal mean matching). *Let $\mathcal{G}$ be a DAG and $\mathcal{I}^*$ be the unique solution to the causal mean matching problem with desired mean $\mu^*$. Algorithm 4 that takes $\mathcal{E}(\mathcal{G})$ and $\mu^*$ as input, runs in polynomial time and adaptively intervenes on set $\mathcal{I} \subseteq V$, identifies $\mathcal{I}^*$ with $\mathbb{E}[|\mathcal{I}|] \leq \mathcal{O}\left(\log n \cdot \log \omega(\mathcal{G})\right) \cdot |\mathcal{I}^*|$.*

Note that for causal mean matching with unique solution $\mathcal{I}^*$, $|\mathcal{I}^*|$ provides a trivial lower bound on the number of interventions required to match the mean. Therefore this result shows that our algorithm achieves a logarithmic approximation to the optimal required number of interventions.

Of note, this is the *first average-case* competitive algorithm with respect to the instance-based lower bound $|\mathcal{I}^*|$. Prior to our work, [ZSU21] provided an efficient algorithm, which is $\log(n)$-competitive with respect to any algorithm in the *worst case*. Such worst-case analysis is equivalent to the scenario where there exists an adaptive adversary when running the algorithm. This may be limited as it relies on extreme cases that may not accurately represent real-world scenarios. Furthermore, the number of interventions required by any algorithm in the worst case can be much larger than the instance-based lower bound $|\mathcal{I}^*|$. For example, if $\mathcal{G}$ is a clique and $\mathcal{I}^*$ is an atomic intervention on its source vertex, then $|\mathcal{I}^*|$ is 1 but the number of interventions required by any algorithm in the worst case is $n$.[11]

## 4 Algorithm for Meek Separator

For any general DAG $\mathcal{G}$, we consider the largest connected component $\mathcal{H} \in CC(\mathcal{E}(\mathcal{G}))$. It is well known that $\mathcal{H}$ is a moral DAG. If $\mathcal{I}$ is a $1/2$-Meek separator for $\mathcal{H}$, we show that $\mathcal{I}$ also serves as a $1/2$-Meek separator for $\mathcal{G}$ (Appendix D). Therefore, for the remainder of the section, we make the assumption that $\mathcal{G}$ is a moral DAG without loss of generality.

### 4.1 Existence of Size-2 Meek Separator

For any vertex $v \in V(\mathcal{G})$, let $A_v = V(\mathcal{G}) \setminus \texttt{Des}[v]$ and $B_v = \texttt{Des}(v)$. Note that $v \notin A_v$ and $v \notin B_v$, therefore $|A_v| + |B_v| + 1 = |V(\mathcal{G})|$. At the heart of our algorithm is the following result which shows the existence of a Meek separator of size at most 2 with some nice properties.

**Lemma 10** (Meek separator). *Let $\mathcal{G}$ be a moral DAG and $K$ be a $1/2$-clique separator of $\mathcal{G}$. There exists a vertex $u \in K$ satisfying the constraints $|A_u| \leq |V(\mathcal{G})|/2$ and $|A_v| > |V(\mathcal{G})|/2$ for all $v \in \texttt{Des}(u) \cap K$. Furthermore, such a vertex $u$ satisfies one of the two conditions: 1). either $u$ is a sink*

---

[10]$v$ is a source vertex if and only if it has no parents.

[11]This can be deduced by using Lemma 5 in [ZSU21].

*vertex*[12] *of* $\mathcal{G}[K]$, *or 2). there exists a vertex* $x$ *such that,* $x \in \mathtt{Des}(u) \cap K$ *and* $\big((V(\mathcal{G}) \backslash \mathtt{Des}[x]) \cap \mathtt{Des}(u)\big) \cap K = \varnothing$ *(i.e.,* $x$ *and* $u$ *are consecutive vertices in the valid permutation of clique* $K$). *In both the cases respectively, either* $\{\{u\}\}$ *or* $\{\{u\}, \{x\}\}$ *is a* 1/2-*Meek separator.*

**Proof Sketch.** Here we present an overview of the proof, focusing solely on the case where $u$ is not a sink node of $K$, since this case encompasses all the key concepts. To demonstrate the existence (Lemma 10), we begin by establishing the following crucial result.

**Lemma 11** (Connected components). *Let* $\mathcal{G}$ *be a moral DAG and* $v \in V(\mathcal{G})$ *be an arbitrary vertex. Any connected component* $\mathcal{H} \in CC(\mathcal{E}_v(\mathcal{G}))$ *satisfies one of the following conditions:* $V(\mathcal{H}) = v$ *or* $V(\mathcal{H}) \subseteq B_v$ *or* $V(\mathcal{H}) \subseteq A_v$.

Returning to Lemma 10, consider vertices $u$ and $x$ and note that when we intervene on both $u$ and $x$, all the connected components within $CC(\mathcal{E}_{\{u,x\}}(\mathcal{G}))$ are either individual nodes or subsets of the subgraph induced by either $A_u$ or $B_x$ or $B_u \cap A_x$. Since $|A_u| \leq {}^{|V(\mathcal{G})|}/_2$ and $|B_x| \leq {}^{|V(\mathcal{G})|}/_2$ (as $|A_x| > {}^{|V(\mathcal{G})|}/_2$), we can conclude that all the connected components within $A_u$ or $B_x$ have a maximum size of $|V(\mathcal{G})|/2$. Therefore, all that remains now is to bound the size of connected components that are subsets of $B_u \cap A_x$. Notably, these connected components $\mathcal{H}$ have no intersection with the clique $K$ and satisfy the condition $V(\mathcal{H}) \cap V(K) = \varnothing$. To establish a size bound for these connected components, we prove the following result.

**Lemma 12** (Size of connected components). *Let* $\mathcal{G}$ *be a graph and* $K$ *be an* $\alpha$-*separator of* $\mathcal{G}$. *Suppose* $\mathcal{H}$ *is a connected subgraph of* $\mathcal{G}$ *and* $V(\mathcal{H}) \cap K = \varnothing$, *then* $|V(\mathcal{H})| \leq \alpha \cdot |V(\mathcal{G})|$.

Combining all the previous analyses, we conclude that all the connected components within $CC(\mathcal{E}_{\{u,x\}}(\mathcal{G}))$ have a maximum size of $|V(\mathcal{G})|/2$, and the set $\{u, x\}$ is a 1/2-Meek separator.

## 4.2 Binary Search Algorithm

To make the existence result algorithmic, we additionally observe the following structural properties.

**Lemma 13** (Properties of connected components). *Consider a moral DAG* $\mathcal{G}$ *and let* $K$ *be an* $\alpha$-*clique separator. Let* $v_1, v_2, \ldots, v_k$ *be the vertices of this clique in a valid permutation. We observe that* $|B_{v_{i+1}}| \leq |B_{v_i}|$, *which in turn implies* $|A_{v_{i+1}}| \geq |A_{v_i}|$. *Additionally, we have* $|A_{v_1}| \leq \alpha \cdot |V(\mathcal{G})|$.

To identify the Meek separator set $\{u, x\}$, we simply need to locate two consecutive vertices in the clique where $|A_u| \leq {}^{|V(\mathcal{G})|}/_2$ and $|A_x| > {}^{|V(\mathcal{G})|}/_2$. The aforementioned result indicates that we can utilize standard randomized binary search procedures to find these vertices efficiently.

---

**Algorithm 1** MeekSeparator$(\mathcal{G}, {}^1/_2)$

---

1: **Input**: Essential graph $\mathcal{E}(\mathcal{G})$ of a moral DAG $\mathcal{G}$.
2: **Output**: A 1/2-Meek separator $\mathcal{I} \cup \mathcal{J}$.
3: Let $K$ be a 1/2-clique separator (Lemma 2). Initialize $i = 0$, $K_0 = K$ and $\mathcal{I}_0 = \varnothing$.
4: **while** $K_i$ is non empty **do**
5:     Let $u_i$ be a uniform random vertex from $K_i$. Intervene on $u_i$.
6:     Find $\mathcal{H}_i \in CC(\mathcal{E}_{u_i}(\mathcal{G}))$ such that $|V(\mathcal{H}_i)|$ is maximized.
7:     **if** $|V(\mathcal{H}_i)| \leq {}^{|V(\mathcal{G})|}/_2$ **then**                        ▷ $u_i$ is a 1/2-Meek separator.
8:         Let $u = u_i$, $\mathcal{I} = \cup_{j=1}^i \{\{u_j\}\}$, and **return** $\mathcal{I}$.
9:     Let $P_{u_i} = \{v \in K \mid v \to u_i\}$ and $Q_{u_i} = \{v \in K \mid v \leftarrow u_i\}$.
10:     **if** there exists a directed path from $u_i$ to $\mathcal{H}_i$ **then**    ▷ $V(\mathcal{H}_i) \subseteq \mathtt{Des}(u_i)$ and $|A_{u_i}| \leq {}^{|V(\mathcal{G})|}/_2$
11:         Update $K_{i+1} = K_i \cap Q_{u_i}$ and $u = u_i$.
12:     **else**
13:         Update $K_{i+1} = K_i \cap P_{u_i}$ and $x = u_i$.
14:     Update $i \leftarrow i + 1$.
15: Let $\mathcal{I} = \{\{x\}\} \cup_{j=1}^{i-1} \{\{u_j\}\}$.
16: **return** $\mathcal{I}$.

---

Figure 4 gives an example of the proposed Algorithm.[13] In general, at each iteration $i$, we randomly select a vertex $u_i$ and determine the connected component $\mathcal{H}_i$ with the maximum cardinality. If

---

[12]$u$ is a sink vertex if and only if it has no children.

[13]In Appendix C, we provide an extended example where the skeleton is not complete, highlighting that the Meek separator can be found by focusing on the 1/2-clique separator.

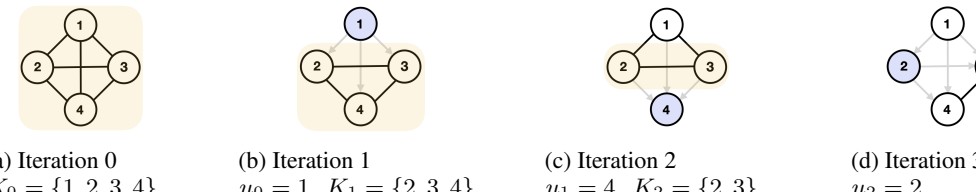

(a) Iteration 0
$K_0 = \{1, 2, 3, 4\}$.

(b) Iteration 1
$u_0 = 1$, $K_1 = \{2, 3, 4\}$.

(c) Iteration 2
$u_1 = 4$, $K_2 = \{2, 3\}$.

(d) Iteration 3
$u_2 = 2$.

Figure 4: An example of Algorithm 1 finding the Meek separator in the ground-truth DAG in Fig. 3a. The sets $K_i$ are highlighted; oriented edges in $\mathcal{E}_{u_i}(\mathcal{G})$ by intervening on $u_i$ are in grey; connected components in $CC(\mathcal{E}_{u_i}(\mathcal{G}))$ are in black. **(a)** suppose we take $K_0 = K = V(\mathcal{G})$ as the $1/2$-clique separator. **(b)** suppose we pick $u_0 = 1$ from $K_0$, then $K_1 = K_0 \cap Q_{u_0} = \{2, 3, 4\}$. **(c)** suppose we pick $u_1 = 4$ from $K_1$, then $K_2 = K_1 \cap P_{u_1} = \{2, 3\}$. **(d)** suppose we pick $u_2 = 2$ from $K_2$, then Algorithm 1 terminates (line 7) returning meek separator $\mathcal{I} = \{2\}$ and $\mathcal{J} = \{1, 4\}$ that helps find it.

$|V(\mathcal{H}_i)| \leq |V(\mathcal{G})|/2$, we have found a Meek separator. Otherwise, we verify if a directed path exists from $u_i$ to $\mathcal{H}_i$. In appendix (Lemma 19), we demonstrate that if such a path exists, then $V(\mathcal{H}_i) \subseteq \text{Des}(u_i) = B_{u_i}$, implying that $|A_{u_i}| \leq |V(\mathcal{G})|/2$ (since $|A_{u_i}| + |B_{u_i}| + 1 = |V(\mathcal{G})|$). To confirm if $u_i$ is the desired vertex $u$, we need to verify that all its descendants satisfy $|A_v| > |V(\mathcal{G})|/2$. Consequently, we recursively examine the vertices in $K$ that are descendants of $u_i$ in $K_i$. Similarly, if the $u_i$ to $H_i$ path doesn't exist, then Lemma 19 implies that $V(\mathcal{H}_i) \subseteq V(\mathcal{G})\backslash\text{Des}[u_i] = A_{u_i}$ and $|A_{u_i}| > |V(\mathcal{G})|/2$. Therefore, to find the desired vertex $u$, we recursively examine the vertices in $K$ that are ancestors of $u_i$ in $K_i$. A similar analysis holds for the desired vertex $x$, and intuitively we conclude that the above algorithm outputs an intervention set that contains vertices $u$ and $x$ satisfying the conditions of Lemma 10. The guarantees of the above algorithm are summarized below.

**Lemma 14** (Output of MeekSeparator). *The algorithm* MeekSeparator *performs at most* $\mathcal{O}(\log|K|)$ *interventions in expectation and finds a vertex* $u \in K$ *that is either a $1/2$-Meek separator or satisfies the following conditions: $|A_u| \leq |V(\mathcal{G})|/2$ and $|A_v| > |V(\mathcal{G})|/2$ for all $v \in \text{Des}(u) \cap K$.*

All proofs can be found in Appendix D. Combining Lemma 14 and Lemma 10 results in Theorem 5.

## 5 Algorithm for Subset Search

Here we present our algorithm for the subset search problem that proves Theorem 7. Our algorithm is based on the Meek separator subroutine and a description of the algorithm is given in Algorithm 2. In the remainder of this section, we present a proof sketch to establish the guarantees.

---
**Algorithm 2** Atomic adaptive subset search.

---
1: **Input**: Essential graph $\mathcal{E}(\mathcal{G})$ of a DAG $\mathcal{G}$ and a set of target edges $T \subseteq E(\mathcal{G})$.
2: **Output**: Atomic intervention set $\mathcal{I}$ s.t. $T \subseteq R(\mathcal{G}, \mathcal{I})$.
3: Initialize $i = 0$ and $\mathcal{I}_0 = \varnothing$.
4: **while** $T\backslash R(\mathcal{G}, \mathcal{I}_i) \neq \varnothing$ **do**
5:     Initialize $\mathcal{J}_i \leftarrow \varnothing$
6:     **for** $\mathcal{H} \in CC(\mathcal{E}_{\mathcal{I}_i}(\mathcal{G}))$ of size $|\mathcal{H}| \geq 2$ and $E(\mathcal{H}) \cap T \neq \varnothing$ **do**
7:         Invoke $\mathcal{J}_\mathcal{H} = \text{MeekSeparator}(\mathcal{H}, 1//2)$. Update $\mathcal{J}_i = \mathcal{J}_i \cup \mathcal{J}_\mathcal{H}$.
8:     Update $\mathcal{I}_{i+1} \leftarrow \mathcal{I}_i \cup \mathcal{J}_i$ and $i \leftarrow i + 1$.
9: **return** $\mathcal{I}_i$.

---

*Correctness.* Upon the termination of our algorithm, it is worth noting that all the connected components that encompass the target edges $T$ have a size of 1. This observation leads to the immediate implication that all the edges belonging to $T$ are oriented.

*Competitive Ratio.* To bound the competitive ratio, we first bound the total cost of our algorithm and then relate it to the subset verification number. To bound the total cost, we first show that the number of outer loops in our algorithm is at most $\mathcal{O}(\log n)$. This holds because, in each outer loop, the size of the connected components containing the target edges $T$ decreases at least by a multiplicative factor of $1/2$. After $\mathcal{O}(\log n)$ such loops, all these connected components will have a size of 1, implying that all edges in $T$ are successfully oriented, leading to the termination of our algorithm.

Next, to bound the cost per outer loop, we consider $\mathcal{J}_i$, which represents the set of interventions performed during the $i$-th loop. It is worth noting that $\mathcal{J}_i$ is a union of Meek separators $\mathcal{J}_\mathcal{H}$ for

each connected component $\mathcal{H} \in CC(\mathcal{E}_{\mathcal{I}_i}(\mathcal{G}))$ such that $E(\mathcal{H}) \cap T \neq \varnothing$. Thus, the cost per loop can be expressed as $|\mathcal{J}_i| = \sum_{\mathcal{H} \in CC(\mathcal{E}_{\mathcal{I}_i}(\mathcal{G}))} \mathbf{1}(E(\mathcal{H}) \cap T \neq \varnothing) \cdot |\mathcal{J}_{\mathcal{H}}|$. By Theorem 5, the Meek separator we obtained for each $\mathcal{H}$ is of size $\mathcal{O}(\log \omega(\mathcal{H})) \leq \mathcal{O}(\log \omega(\mathcal{G}))$. We can conclude that the cost per iteration is at most: $|\mathcal{J}_i| \leq \mathcal{O}(\log \omega(G)) \sum_{\mathcal{H} \in CC(\mathcal{E}_{\mathcal{I}_i}(\mathcal{G}))} \mathbf{1}(E(\mathcal{H}) \cap T \neq \varnothing)$. To relate this cost to the subset verification number, we utilize the lower bound result from Lemma 6, which states that: $\nu_1(\mathcal{G}, T) \geq \max_{\mathcal{I} \subseteq V} \sum_{\mathcal{H} \in CC(\mathcal{E}_{\mathcal{I}}(\mathcal{G}))} \mathbf{1}(E(\mathcal{H}) \cap T \neq \varnothing)$. Combining the above equations, we obtain $|\mathcal{J}_i| \leq \mathcal{O}(\log \omega(\mathcal{G}))\nu_1(\mathcal{G}, T)$. As there are at most $\mathcal{O}(\log n)$ iterations in total, the total cost of our algorithm can be bounded by: $\mathcal{O}(\log n \cdot \log \omega(\mathcal{G}))\nu_1(\mathcal{G}, T)$. We defer the detailed proofs of Theorem 7 (upper bound) and Lemma 6 (lower bound) to Appendix E.

## 6 Algorithm for Causal Mean Matching

Here we study the causal mean matching problem. En route, we provide an algorithm that, given an essential graph $\mathcal{E}(\mathcal{G})$ of a DAG $\mathcal{G}$ and a subset $U$ of vertices, finds a source vertex within the induced subgraph $\mathcal{G}[U]$. This source vertex, denoted as $s$, satisfies the property that there exists no vertex $v \in U$ such that $v \in \mathtt{Anc}(s)$. The description is given in Algorithm 3. This algorithm identifies a source vertex of $U$ in $\mathcal{O}(\log n \cdot \log \omega(\mathcal{G}))$ interventions and its guarantees are summarized in Lemma 8. Below, we present a concise overview of its proof.

---

**Algorithm 3** FindSource($\mathcal{E}(\mathcal{G}), U$)

---

1: **Input**: Essential graph $\mathcal{E}(\mathcal{G})$ of a DAG $\mathcal{G}$ and a subset of vertices $U \subseteq V(\mathcal{G})$.
2: **Output**: Atomic intervention set $\mathcal{I}$ and a source vertex in $U$.
3: Initialize $i = 0$ and $\mathcal{I}_0 = \varnothing$.
4: Let $C_0 = \{\mathcal{H} \in CC(\mathcal{E}_{\mathcal{I}_0}(\mathcal{G})) \mid V(\mathcal{H}) \cap U \neq \varnothing\}$ and let $\mathcal{H}_0 \in C$ be a connected component with no incoming directed path from any other component $\mathcal{H} \in C_0$.
5: **while** $|V(\mathcal{H}_i) \cap U| > 1$ **do**
6:     Compute $\mathcal{J}_i = \text{MeekSeparator}(\mathcal{E}(\mathcal{H}_i), 1/2)$ and intervene on it.
7:     Let $C_{i+1} = \{\mathcal{H} \in CC(\mathcal{E}_{\mathcal{J}_i}(\mathcal{H}_i)) \mid V(\mathcal{H}) \cap U \neq \varnothing\}$ and let $\mathcal{H}_{i+1} \in C_{i+1}$ be a connected component with no incoming directed path from any other component $\mathcal{H} \in C_{i+1}$.
8:     Let $\mathcal{I}_{i+1} = \mathcal{I}_i \cup \mathcal{J}_i$. Increment $i$ by 1.
9: **return** $\mathcal{I}_i$ and $V(\mathcal{H}_i)$.

---

As stated in the description, our algorithm invokes the Meek separator in each iteration and identifies the connected component containing a source vertex $s$ and recurses on it. This connected component can be identified by finding the component that has no incoming directed path from any other components. Then as we invoke the Meek separator in each iteration, the size of the connected component decreases at least by a factor of $1/2$. Therefore the algorithm terminates in $\mathcal{O}(\log n)$. Since we perform at most $\mathcal{O}(\log \omega(\mathcal{G}))$ interventions in each iteration, the total number of interventions performed by our algorithm is at most $\mathcal{O}(\log n \cdot \log \omega(\mathcal{G}))$. This concludes our proof overview for the source-finding algorithm. In the remainder, we use it to solve the causal mean matching problem.

*Causal Mean Matching.* We consider the same setting as in [ZSU21] with atomic interventions, where the goal is to find a set of shift interventions $\mathcal{I}$ such that the mean of the interventional distribution $\mathbb{E}_{P^{\mathcal{I}}}[V]$ matches a desired mean $\mu^*$. An atomic shift intervention set $\mathcal{I}$ with shift values $\{a_i\}_{i \in \mathcal{I}}$ modifies the conditional distribution as $P^{\mathcal{I}}(v_i = x + a_i \mid v_{\mathtt{Pa}(i)}) = P(v_i = x \mid v_{\mathtt{Pa}(i)})$ for $i \in \mathcal{I}$. In particular, [ZSU21] show that there exists a unique solution $\mathcal{I}^*$ such that $\mathbb{E}_{P^{\mathcal{I}^*}}[V] = \mu^*$ and to find such $\mathcal{I}^*$, it suffices to iteratively find the source vertices of all vertices whose means differ from that of $\mu^*$. The intuition behind this is that (1) intervening on other vertices will not change the mean of the source vertex, and (2) the shift value of the source vertex equals exactly the mean discrepancy with respect to $\mu^*$. Thus, our Algorithm 3 can be used as a subroutine to solve for $\mathcal{I}^*$ iteratively. We describe the full procedure in Algorithm 4 in Appendix F. Our analysis of Lemma 8 is used to derive the guarantee of Algorithm 4 in Theorem 9. Details are deferred to Appendix F.

## 7 Experiments

Here, we implement our Meek separator to solve for subset search and causal mean matching discussed in the previous sections. Details and extended experiments are provided in Appendix G.

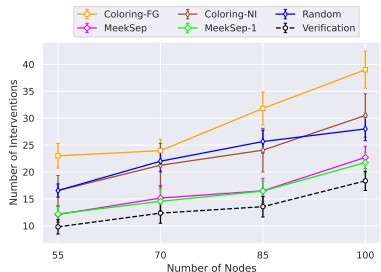
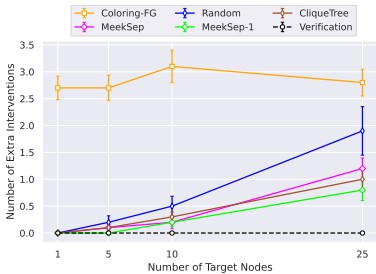

(a) Subset search on $r$-hop model with $r = 3$     (b) Causal mean matching with varying $|\mathcal{I}^*|$

Figure 5: Meek separator for **(a)** subset search and **(b)** causal matching. Each dot is averaged across 20 DAGs, where the error bar shows $0.5$ and $0.2$ standard deviation in (a) and (b), respectively.

**Subset Search.** For this experiment, we consider the local causal graph discovery problem where the goal is to identify the target edges within an $r$-hop neighborhood of a random vertex $v$. We compare our method in Algorithm 2 and its variant, `MeekSep` and `MeekSep-1`, against four carefully constructed baselines. The variant `MeekSep-1` runs Algorithm 2 but checks after performing every intervention inside line 7 and terminates if the subset search problem is solved. The `Random` baseline intervenes on a randomly sampled vertex at every step and terminates when the subset search problem is solved. The `Coloring-FG` baseline identifies the full causal graph using [SKDV15], where `Coloring-NI` is the variant that only identifies the subgraph induced by vertices incident to the target edges, similar to the method suggested by [CS23]. Our proposed methods consistently outperform these baselines across different graph sizes in Figure 5a. Finally, `Verification` shows the subset search verification number [CS23] which serves as a lower bound.

**Causal Mean Matching.** We consider Erdös-Rényi graphs [ER60] with 50 vertices where the ground-truth solution $\mathcal{I}^*$ is randomly sampled from these vertices. We compare our Algorithm 4 and its variant, `MeekSep` and `MeekSep-1`, against four baselines proposed in [ZSU21]. The variant is in the fashion described for the subset search. The `Random` and `CliqueTree` baselines use the same backbone as `MeekSep`, but search for source vertex using randomly sampled interventions and the clique-tree strategy proposed in [ZSU21], respectively. `Coloring-FG` first identifies the full graph then solves for $\mathcal{I}^*$, whereas `Verification` is the lower bound that uses $|\mathcal{I}^*|$ interventions. The number of extra interventions relative to `Verification` is shown in Figure 5b, where we observe our methods to outperform `Random` and `Coloring-FG`. Empirically, our approach is competitive with the state-of-the-art method `CliqueTree` while providing far better theoretical guarantees.

# 8 Discussion

In this work, we introduced Meek separators. In particular, we established the existence of a small-sized Meek separator and presented efficient algorithms to compute it. Meek separators hold great potential for designing divide-and-conquer strategies to tackle various causal discovery problems. We demonstrated this by designing efficient approximation algorithms for two important problems in targeted causal discovery: subset search and causal mean matching. Our approximation guarantees are exponentially better than the guarantees achievable by any deterministic algorithm for both problems. It would be an interesting future research endeavour to explore the application of Meek separators to address other problems in the field of causal discovery.

**Limitations and Future Work.** We made several standard assumptions such as causal sufficiency and we considered the noiseless setting. In future work, it would be of interest to relax some of these assumptions. Particularly, we believe that investigating the sample complexity for conducting targeted causal discovery would be an important avenue to pursue.

In addition to these broad questions, there are some specific open problems. One such problem is understanding the weighted subset search problem. Although efficient algorithms have been proposed in [CS23] to compute weighted subset verification numbers, the weighted subset search problem remains open. Exploring its approximability would be an interesting research direction. Moreover, for the causal mean matching problem, extending the matching criteria beyond the mean to encompass other higher-order moments of the distribution would be a natural and compelling future direction.

## Acknowledgements

The authors were supported by the the Eric and Wendy Schmidt Center at the Broad Institute, as well as NCCIH/NIH (1DP2AT012345), ONR (N00014-22-1-2116), DOE-ASCR (DE-SC0023187), the MIT-IBM Watson AI Lab, and a Simons Investigator Award. J.Z. was partially supported by an Apple AI/ML PhD Fellowship.

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

# Contents of Appendix

# A Meek Rules

Meek rules refer to a collection of four edge orientation rules that are proven to be sound and complete when applied to a set of arcs that possesses a consistent extension to a directed acyclic graph (DAG) [Mee95]. With the presence of edge orientation information, it is possible to iteratively apply Meek rules until reaching a fixed point, thereby maximizing the number of oriented arcs.

**Definition 15** (Consistent extension). For a given graph $G$, a set of arcs is considered to have a *consistent DAG extension* $\pi$ if there exists a permutation of the vertices satisfying the following conditions: (i) for every edge $\{u, v\}$ in $G$, it is oriented as $u \to v$ whenever $\pi(u) < \pi(v)$, (ii) there are no directed cycles, and (iii) all the given arcs are included in the extension.

**Definition 16** (The four Meek rules [Mee95], see Figure 6 for an illustration).

**R1** Edge $\{a, b\} \in E \setminus A$ is oriented as $a \to b$ if $\exists\, c \in V$ such that $c \to a$ and $c \not\sim b$.

**R2** Edge $\{a, b\} \in E \setminus A$ is oriented as $a \to b$ if $\exists\, c \in V$ such that $a \to c \to b$.

**R3** Edge $\{a, b\} \in E \setminus A$ is oriented as $a \to b$ if $\exists\, c, d \in V$ such that $d \sim a \sim c$, $d \to b \leftarrow c$, and $c \not\sim d$.

**R4** Edge $\{a, b\} \in E \setminus A$ is oriented as $a \to b$ if $\exists\, c, d \in V$ such that $d \sim a \sim c$, $d \to c \to b$, and $b \not\sim d$.

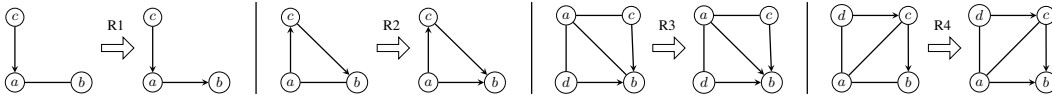

Figure 6: An illustration of the four Meek rules

An algorithm [WBL21, Algorithm 2] has been developed to compute the closure under Meek rules efficiently. The algorithm runs in $\mathcal{O}(d \cdot |E|)$ time, where $d$ represents the degeneracy of the graph skeleton[14].

# B Preliminaries and Other Useful Results

Here we state some useful notation and results. For an arc $u \to v$ in $\mathcal{G}$, we define $R_1^{-1}(\mathcal{G}, u \to v) \subseteq V$ and $R_k^{-1}(\mathcal{G}, u \to v) \subseteq 2^V$ to refer to interventions orienting an arc $u \to v$. Equivalently, $R_1^{-1}(\mathcal{G}, u \to v) = \{w \in V : u \to v \in R(\mathcal{G}, w)\}$ and $R_k^{-1}(\mathcal{G}, u \to v) = \{I \subseteq V : |I| \leq k, u \to v \in R(\mathcal{G}, I)\}$.

**Proposition 17** (Theorem 7 in [CS23]). *Consider a DAG $\mathcal{G} = (V, E)$ and intervention sets $\mathcal{I}, \mathcal{J} \subseteq 2^V$. The following statements are true: 1) $skel(\mathcal{G}^{\mathcal{I}})$ is exactly the chain components of $\mathcal{E}_{\mathcal{I}}(\mathcal{G})$. 2) $\mathcal{G}^{\mathcal{I}}$ does not have new v-structures. 3) For any two vertices $u$ and $v$ in the same chain component of $\mathcal{E}_{\mathcal{I}}(\mathcal{G})$, we have $Pa_{\mathcal{G},\mathcal{I}}(u) = Pa_{\mathcal{G},\mathcal{I}}(v)$. 4) If $u \to v \in R(\mathcal{G}, \mathcal{I})$, then $u$ and $v$ belong to different chain components of $\mathcal{E}_{\mathcal{I}}(\mathcal{G})$. 5) Any acyclic completion of $\mathcal{E}(\mathcal{G}^{\mathcal{I}})$ can be combined with $R(\mathcal{G}, \mathcal{I})$ to obtain a valid DAG that has the same essential graph and $\mathcal{I}$-essential graph as $\mathcal{E}(\mathcal{G})$ and $\mathcal{E}_{\mathcal{I}}(\mathcal{G})$, respectively. 6) $R(\mathcal{G}^{\mathcal{I}}, \mathcal{J}) = R(\mathcal{G}, \mathcal{J}) \setminus R(\mathcal{G}, \mathcal{I})$. 7) $R(\mathcal{G}, \mathcal{I} \cup \mathcal{J}) = R(\mathcal{G}^{\mathcal{I}}, \mathcal{J}) \,\dot{\cup}\, R(\mathcal{G}, \mathcal{I})$. 8) $R(\mathcal{G}, \mathcal{I} \cup \mathcal{J}) = R(\mathcal{G}^{\mathcal{I}}, \mathcal{J}) \,\dot{\cup}\, R(\mathcal{G}^{\mathcal{J}}, \mathcal{I}) \,\dot{\cup}\, \big(R(\mathcal{G}, \mathcal{I}) \cap R(\mathcal{G}, \mathcal{J})\big)$.*

**Lemma 18** (Theorem 10 in [CS23]). *Let $\mathcal{G} = (V, E)$ be a DAG without v-structures and $u \to v$ in $\mathcal{G}$ be unoriented in $\mathcal{E}(\mathcal{G})$. Then, $R_1^{-1}(\mathcal{G}, u \to v) = Des[w] \cap Anc[v]$ for some $w \in Anc[u]$.*

# C Another Example of Algorithm 1

We provide another example of Algorithm 1 in an incomplete graph, highlighting that Meek separator by solely focusing on the $1/2$-clique separator.

---

[14]A $d$-degenerate graph is an undirected graph in which every subgraph has a vertex of degree at most $d$. Note that the degeneracy of a graph is typically smaller than the maximum degree of the graph.

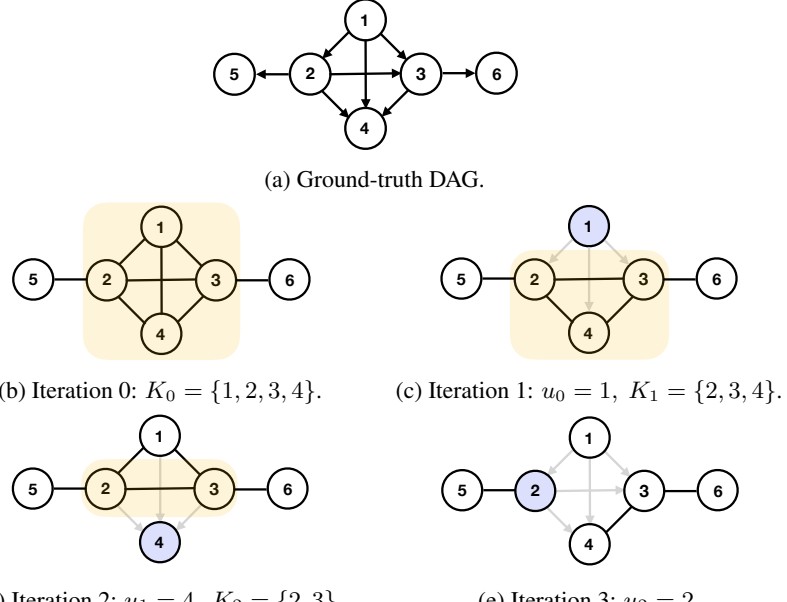

(a) Ground-truth DAG.

(b) Iteration 0: $K_0 = \{1, 2, 3, 4\}$.    (c) Iteration 1: $u_0 = 1$, $K_1 = \{2, 3, 4\}$.

(d) Iteration 2: $u_1 = 4$, $K_2 = \{2, 3\}$.    (e) Iteration 3: $u_2 = 2$.

Figure 7: An example of Algorithm 1 finding the Meek separator in an incomplete graph. The sets $K_i$ are highlighted; oriented edges in $\mathcal{E}_{u_i}(\mathcal{G})$ by intervening on $u_i$ are in grey; connected components in $CC(\mathcal{E}_{u_i}(\mathcal{G}))$ are in black. **(a)** ground-truth DAG $\mathcal{G}$. **(b)** suppose we take $K_0 = K = V(\mathcal{G})$ as the $^1/_2$-clique separator. **(c)** suppose we pick $u_0 = 1$ from $K_0$, then $K_1 = K_0 \cap Q_{u_0} = \{2, 3, 4\}$. **(d)** suppose we pick $u_1 = 4$ from $K_1$, then $K_2 = K_1 \cap P_{u_1} = \{2, 3\}$. **(e)** suppose we pick $u_2 = 2$ from $K_2$, then Algorithm 1 terminates (line 7) returning meek separator $\mathcal{I} = \{2\}$ and $\mathcal{J} = \{1, 4\}$ that helps find it.

# D   Remaining Proofs for Meek Separator

Here we provide all the remaining proofs for the Meek separator algorithm.

## D.1   Proof for Lemma 11

**Lemma 11** (Connected components). *Let $\mathcal{G}$ be a moral DAG and $v \in V(\mathcal{G})$ be an arbitrary vertex. Any connected component $\mathcal{H} \in CC(\mathcal{E}_v(\mathcal{G}))$ satisfies one of the following conditions: $V(\mathcal{H}) = v$ or $V(\mathcal{H}) \subseteq B_v$ or $V(\mathcal{H}) \subseteq A_v$ .*

*Proof.* Performing an intervention on node $v$ results in orienting all its adjacent edges. Consequently, one of the connected components in $CC(\mathcal{E}_v(\mathcal{G}))$ is $\{v\}$, and it is adequate to focus on the remaining connected components.

Suppose, for the sake of contradiction, that there exists a connected component $\mathcal{H} \in CC(\mathcal{E}_v(\mathcal{G}))$ such that $\mathcal{H}$ contains two vertices $a$ and $b$, such that, $a \in A_v$ and $b \in B_v$. Since $a$ and $b$ belong to the same connected component, we consider the path within $H$ that connects these vertices. Notably, this path includes two adjacent vertices $c$ and $d$, where $c \in A_v$ and $d \in B_v$, and the edge $(c, d)$ remains unoriented.

However, according to Lemma 18, intervening on any vertex within the set $\mathtt{Des}[w] \cap \mathtt{Anc}[d]$ for some fixed $w \in \mathtt{Anc}[c]$ will orient edge $(c, d)$. As $w \in \mathtt{Anc}[c]$, we have $\mathtt{Des}[c] \subseteq \mathtt{Des}[w]$ and therefore $\mathtt{Des}[c] \cap \mathtt{Anc}[d] \subseteq \mathtt{Des}[w] \cap \mathtt{Anc}[d]$. Consequently as $v \in \mathtt{Des}[c] \cap \mathtt{Anc}[d]$, we get that intervening on $v$ orients the edge $(c, d)$.

Hence, it is impossible for vertices $a \in A_v$ and $b \in B_v$ to belong to the same connected component. Consequently, we can conclude the proof. □

## D.2 Proof for Lemma 12

**Lemma 12** (Size of connected components). *Let $\mathcal{G}$ be a graph and $K$ be an $\alpha$-separator of $\mathcal{G}$. Suppose $\mathcal{H}$ is a connected subgraph of $\mathcal{G}$ and $V(\mathcal{H}) \cap K = \varnothing$, then $|V(\mathcal{H})| \leq \alpha \cdot |V(\mathcal{G})|$.*

*Proof.* Since $\mathcal{H}$ is a connected subgraph of $\mathcal{G}$ and $V(\mathcal{H}) \cap V(K) = \varnothing$, removing $K$ from $\mathcal{G}$ does not result in the deletion of any edges or vertices from the subgraph $\mathcal{H}$. Consequently, the vertices in $\mathcal{H}$ will remain connected even after the removal of $K$, and they will all belong to the same connected component in the graph $\mathcal{G}\backslash K$. Considering that $K$ is an $\alpha$-separator, this further implies that $|V(\mathcal{H})| \leq \alpha \cdot |V(\mathcal{G})|$. Thus, we can conclude the proof. $\square$

## D.3 Proof for Lemma 13

**Lemma 13** (Properties of connected components). *Consider a moral DAG $\mathcal{G}$ and let $K$ be an $\alpha$-clique separator. Let $v_1, v_2, \ldots, v_k$ be the vertices of this clique in a valid permutation. We observe that $|B_{v_{i+1}}| \leq |B_{v_i}|$, which in turn implies $|A_{v_{i+1}}| \geq |A_{v_i}|$. Additionally, we have $|A_{v_1}| \leq \alpha \cdot |V(\mathcal{G})|$.*

*Proof.* Since $v_i$ precedes $v_{i+1}$ in the true ordering defined by the underlying ground truth DAG $\mathcal{G}$, it follows that $v_{i+1} \in \texttt{Des}(v_i)$. Consequently, we can deduce that $\texttt{Des}(v_{i+1}) \subseteq \texttt{Des}(v_i)$, which implies $|B_{v_{i+1}}| \leq |B_{v_i}|$. Additionally, note that $|A_v| + |B_v| + 1 = |V(\mathcal{G})|$. Therefore, it holds that $|A_{v_{i+1}}| \geq |A_{v_i}|$.

Furthermore, as $v_1$ is the source node of the clique, removing $K$ ensures that all vertices in $A_{v_1}$ are still in the remaining graph. They also induce a connected subgraph. Otherwise suppose $a \in A_{v_1}$ and $b \in A_{v_1}$ are not connected. Then the parent $c$ of $v_1$ on a path from $a$ to $v_1$ and the parent $d$ of $v_1$ on a path from $b$ to $v_1$ are not connected. This creates a v-structure $c \rightarrow v_1 \leftarrow d$, contradicting $\mathcal{G}$ being moral. Thus $A_{v_1}$ is a connected subgraph with no intersection with $K$. As $K$ is a $\alpha$-clique separator, we have by Lemma 12 that $|A_{v_1}| \leq \alpha \cdot |V(\mathcal{G})|$. We conclude the proof. $\square$

## D.4 Proof for Lemma 10

**Lemma 10** (Meek separator). *Let $\mathcal{G}$ be a moral DAG and $K$ be a $1/2$-clique separator of $\mathcal{G}$. There exists a vertex $u \in K$ satisfying the* constraints *$|A_u| \leq |V(\mathcal{G})|/2$ and $|A_v| > |V(\mathcal{G})|/2$ for all $v \in \texttt{Des}(u) \cap K$. Furthermore, such a vertex $u$ satisfies one of the two* conditions*: 1). either $u$ is a sink vertex[15] of $\mathcal{G}[K]$, or 2). there exists a vertex $x$ such that, $x \in \texttt{Des}(u) \cap K$ and $\big((V(\mathcal{G})\backslash \texttt{Des}[x]) \cap \texttt{Des}(u)\big) \cap K = \varnothing$ (i.e., $x$ and $u$ are consecutive vertices in the valid permutation of clique $K$). In both the cases respectively, either $\{\{u\}\}$ or $\{\{u\}, \{x\}\}$ is a $1/2$-Meek separator.*

*Proof.* Consider the vertices $v_1, \ldots, v_k$ of the clique $K$ in the true ordering defined by the underlying ground truth $\mathcal{G}$. Since $K$ is a 1/2-clique separator, according to Lemma 13, we deduce that $|A_{v_1}| \leq |V(\mathcal{G})|/2$. Let $u$ denote the last vertex, in terms of the true ordering, within the clique $K$ that satisfies $A_u \leq |V(\mathcal{G})|/2$. It is important to note that $u$ fulfills the *constraints* specified in the lemma.

Therefore there exists a vertex $u$ that fulfills these *constraints*. Now we show that any vertex that fulfills these constraints will satisfy one of two *conditions*, and that either $\{\{u\}\}$ or $\{\{u\}, \{x\}\}$ is a $1/2$-Meek separator.

Suppose $u$ is a sink vertex of $K$. Let $\mathcal{H} \in CC(\mathcal{E}_u(\mathcal{G}))$ and note that according to Lemma 11, $V(\mathcal{H}) = \{u\}$ or $V(\mathcal{H}) \subseteq \texttt{Des}(u)$ or $V(\mathcal{H}) \subseteq V(\mathcal{G})\backslash\texttt{Des}[u]$. Since $|A_u| \leq |V(\mathcal{G})|/2$, any $\mathcal{H}$ such that $V(\mathcal{H}) \subseteq V(\mathcal{G})\backslash\texttt{Des}[u]$ satisfies $|V(\mathcal{H})| \leq |V(\mathcal{G})|/2$. Now, suppose $V(\mathcal{H}) \subseteq \texttt{Des}(u)$. In this case, we observe that $V(\mathcal{H}) \cap K = \varnothing$ and $\mathcal{H}$ is a connected subgraph of $G$. By utilizing Lemma 12, we immediately have $|V(\mathcal{H})| \leq |V(\mathcal{G})|/2$. Therefore, $\{\{u\}\}$ will be a 1/2-Meek separator.

Suppose $u$ is not a sink vertex. Consider the vertex $x$ that follows $u$ within the clique $K$ in the ordering defined by the DAG $\mathcal{G}$. Note that $x \in \texttt{Des}(u) \cap K$ and $(\texttt{Des}(u) \cap (V(\mathcal{G})\backslash\texttt{Des}[x])) \cap K = \varnothing$. Furthermore from the definition of $u$, we also have that $|A_x| > |V(\mathcal{G})|/2$, which further implies $|B_x| \leq |V(\mathcal{G})|/2$. As intervening on more vertices only creates finer-grained connected components, we have that for any $\mathcal{J} \subseteq V(\mathcal{G})$, $w \in V(\mathcal{G})$, and $\mathcal{H}' \in CC(\mathcal{E}_{\mathcal{J} \cup \{w\}}(\mathcal{G}))$, there exists an $\mathcal{H}'' \in CC(\mathcal{E}_{\mathcal{J}}(\mathcal{G}))$ such that $\mathcal{H}'$ is a subgraph of $\mathcal{H}''$.

---

[15] $u$ is a sink vertex if and only if it has no children.

Let $\mathcal{I} = \{x, u\}$, consider any $\mathcal{H} \in CC(\mathcal{E}_{\mathcal{I}}(G))$ and note that there exist connected components $M \in CC(\mathcal{E}_u(\mathcal{G}))$ and $L \in CC(\mathcal{E}_x(\mathcal{G}))$ such that $\mathcal{H}$ is a subgraph of both $M$ and $L$. We apply Lemma 11 to deduce that $V(M) = u$ or $V(M) \subseteq \text{Des}(u)$ or $V(M) \subseteq V(\mathcal{G})\backslash\text{Des}[u]$ and $V(L) = x$ or $V(L) \subseteq \text{Des}(x)$ or $V(L) \subseteq V(\mathcal{G})\backslash\text{Des}[x]$.

If $V(M) = u$ or $V(M) \subseteq V(\mathcal{G})\backslash\text{Des}[u]$ or $V(L) = x$ or $V(L) \subseteq \text{Des}(x)$, then $V(\mathcal{H}) = u$ or $V(\mathcal{H}) = x$ or $V(\mathcal{H}) \subseteq A_u$ or $V(\mathcal{H}) \subseteq B_x$. As $|A_u|, |B_x| \le |V(\mathcal{G})|/2$, in all of these cases, we have that $|V(\mathcal{H})| \le |V(\mathcal{G})|/2$. Now consider the remaining cases where $V(M) \subseteq \text{Des}(u)$ and $V(L) \subseteq V(\mathcal{G})\backslash\text{Des}[x]$. As $\mathcal{H}$ is a subgraph of both $M$ and $L$, we have that $\mathcal{H}$ is a subgraph of $C$, where $C$ satisfies $V(C) = \text{Des}(u) \cap (V(\mathcal{G})\backslash\text{Des}[x])$. As $(\text{Des}(u) \cap (V(\mathcal{G})\backslash\text{Des}[x])) \cap K = \varnothing$, we have that $V(C) \cap K = \varnothing$. Furthermore, as $\mathcal{H}$ is a subgraph of $C$ we also have that $V(\mathcal{H}) \cap K = \varnothing$. As $\mathcal{H}$ is a connected subgraph of $G$ and $V(\mathcal{H}) \cap K = \varnothing$, by utilizing Lemma 12, we have that $|V(\mathcal{H})| \le |V(\mathcal{G})|/2$. Therefore in all cases, we have that $|V(\mathcal{H})| \le |V(\mathcal{G})|/2$, and thus $\{\{u\}, \{x\}\}$ is a $1/2$-Meek separator, which concludes the proof. □

## D.5 Proof for Lemma 14

Here we present a proof for Lemma 14, which describes the structure of the solution returned by our Meek separator algorithm. In order to prove this lemma, we first establish the following result.

**Lemma 19.** *Let $\mathcal{G}$ be a connected moral DAG and $v \in V(\mathcal{G})$ be an arbitrary vertex. For any connected component $\mathcal{H} \in CC(\mathcal{E}_v(\mathcal{G}))$, there exists a directed path from $v$ to $\mathcal{H}$ in $\mathcal{E}_v(\mathcal{G})$ if and only if $\mathcal{H}$ satisfies $V(\mathcal{H}) \subseteq \text{Des}(v)$. Furthermore, we can certify if $\mathcal{H}$ is a subset of $\mathcal{G}\backslash\text{Des}[v]$ or $\text{Des}(v)$ in polynomial time.*

*Proof.* Suppose there exists a directed path from $v$ to a connected component $\mathcal{H}$, then note that all the vertices in this directed path are descendants of $v$ and there exists a vertex $u \in V(\mathcal{H})$ such that, $u \in \text{Des}(v)$. Furthermore, by Lemma 11, we know that all the connected components $\mathcal{H}$ should satisfy: $V(\mathcal{H}) = \{v\}$ or $V(\mathcal{H}) = \text{Des}(v)$ or $V(\mathcal{H}) = V(\mathcal{G})\backslash\text{Des}(v)$. Using the fact that there exists a vertex $u \in V(\mathcal{H})$ which belongs to $\text{Des}(v)$. Therefore, $\text{Des}(v) \cap V(\mathcal{H})$ is not empty and we conclude that $V(\mathcal{H}) \subseteq \text{Des}(v)$.

In the subsequent part of the proof, we examine the converse direction. Assume there exists a connected component $\mathcal{H}$ such that $V(\mathcal{H}) \subseteq \text{Des}(v)$, and we aim to demonstrate the existence of a directed path from $v$ to $\mathcal{H}$ in the interventional essential graph $\mathcal{E}_v(\mathcal{G})$. Since $\mathcal{H} \subseteq \text{Des}(v)$, we consider the shortest directed path from $v$ to $\mathcal{H}$ in $\mathcal{G}$. Let $w$ be the endpoint of this path $P$ within $\mathcal{H}$.

Suppose that all edges in $P$ are oriented. In this case, we have already found a directed path from $v$ to $\mathcal{H}$ in $\mathcal{E}_v(\mathcal{G})$, and our objective is achieved. Hence, we focus on the scenario where some edges in $P$ are unoriented. Let $P : v = v_0 \to v_1 \to \cdots \to v_\ell \to w$ denote the vertices along the path $P$, and let $v_j \sim v_{j+1}$ represent the first unoriented edge encountered.

Since all edges incident to $v$ are oriented, we have $v_j \ne v$. Now, consider the situation where $v_{j-1}$ and $v_{j+1}$ are not adjacent. In such a case, the Meek rule R1 would orient the edge $v_j \to v_{j+1}$. However, since $v_j \to v_{j+1}$ remains unoriented, it implies that the edge $v_{j-1} \to v_{j+1}$ must exist. However, this contradicts the fact that $P$ is the shortest directed path. Therefore, it must be the case that $P$ is either entirely oriented or entirely unoriented. As some edges in $P$ are oriented, we conclude that path $P$ is completely oriented, thereby establishing the presence of a directed path from $v$ to $\mathcal{H}$ in the interventional essential graph $\mathcal{E}_v(\mathcal{G})$.

In the remaining part of the proof, we discuss the time complexity of finding these directed paths. To certify whether $V(\mathcal{H}) \subseteq \text{Des}(v)$ or $V(\mathcal{H}) \subseteq V(\mathcal{G})\backslash\text{Des}[v]$, all we need to do is check whether a directed path exists between $v$ and any $u \in V(\mathcal{H})$. If the direction of the path is from $v$ to $\mathcal{H}$, then $V(\mathcal{H}) \subseteq \text{Des}(v)$ or else $V(\mathcal{H}) \subseteq V(\mathcal{G})\backslash\text{Des}[v]$. To check whether a directed path exists between two vertices can be done in polynomial time and therefore we can certify if $V(\mathcal{H}) \subseteq \text{Des}(v)$ or $V(\mathcal{H}) \subseteq V(\mathcal{G})\backslash\text{Des}[v]$. We conclude the proof. □

**Lemma 14** (Output of MeekSeparator)**.** *The algorithm MeekSeparator performs at most $\mathcal{O}(\log |K|)$ interventions in expectation and finds a vertex $u \in K$ that is either a $1/2$-Meek separator or satisfies the following conditions: $|A_u| \le |V(\mathcal{G})|/2$ and $|A_v| > |V(\mathcal{G})|/2$ for all $v \in \text{Des}(u) \cap K$.*

*Proof.* Consider a clique $K$ and a true ordering $\pi$ defined on the vertices $V(\mathcal{G})$ of the underlying ground truth DAG $\mathcal{G}$. Let $v_1, v_2, \ldots, v_k$ represent the vertices in the clique $K$, labeled according to

the true ordering $\pi$. In other words, $v_i$ precedes $v_j$ in $\pi$ whenever $i < j$. Since $K$ is a $1/2$-Clique separator, according to Lemma 10, we have $|A_{v_1}| \leq |V(\mathcal{G})|/2$. Additionally, based on Lemma 13, we know that $|A_{v_j}| \leq |A_{v_{j+1}}|$ for all $j \in [1, k-1]$. Let $v_{j^*}$ be the last vertex within the clique $K$ in the true ordering $\pi$ such that $|A_{v_{j^*}}| \leq |V(\mathcal{G})|/2$.

In our proof, we demonstrate that our algorithm can either discover a $1/2$-Meek separator or identify the vertex $v_{j^*}$ within $\mathcal{O}(\log |K|)$ interventions, on average. If, at any stage of the algorithm, $u_i$ is a $1/2$-Meek separator, our task is complete. Consequently, for the remainder of the proof, we concentrate on the alternative scenario and establish that our algorithm locates $v_{j^*}$. In this case, we observe that either $v_{j^*}$ corresponds to the sink node of $K$ or we uncover the subsequent vertex, $v_{j^*+1}$, in the ordering. Notably, according to Lemma 10, either case implies that $\mathcal{I} = \{u\}$ or $\mathcal{I} = \{u, x\}$ constitutes a $1/2$-Meek separator.

During each iteration $i$ of our algorithm, we intervene on a uniformly random chosen vertex $u_i$ from the remaining clique $K_i$. Let $\mathcal{H}_i$ represent the largest connected component in the interventional essential graph $\mathcal{E}_{u_i}(\mathcal{G})$. The algorithm terminates when the size of the clique becomes empty. Notably, when we intervene on a vertex $u_i$, by Lemma 19, the existence of a directed path from $u_i$ to $\mathcal{H}_i$ implies that $\mathcal{H}_i$ corresponds to a connected component in the descendant subgraph of $u_i$. Since $|V(\mathcal{H}_i)| > |V(\mathcal{G})|/2$, it follows that $|\text{Des}(u_i)| > |V(\mathcal{G})|/2$, which further implies $|A_{u_i}| \leq |V(\mathcal{G})|/2$. We assign $u = u_i$ and observe that $|\text{Des}(u)| = |\text{Des}(u_i)| > |V(\mathcal{G})|/2$, implying $|A_u| \leq |V(\mathcal{G})|/2$. In this scenario, we recursively proceed with the set $Q_{u_i} \cap K_i$, which comprises the nodes that appear after the vertex $u_i$ in the true ordering $\pi$. It is worth noting that these vertices $y \in Q_{u_i} \cap K_i$ satisfy $|A_y| \geq |A_{u_i}|$ since $y \in \text{Des}(u_i)$ (Lemma 10).

In the other case, when no path exists from $u_i$ to $\mathcal{H}_i$, by Lemma 19, we deduce that $V(\mathcal{H}_i) \subseteq A_{u_i}$, thereby leading to $|A_{u_i}| > V(\mathcal{G})/2$. Consequently, we assign $x = u_i$ and therefore $|A_x| > V(\mathcal{G})/2$. In this situation, we recursively proceed with the set $P_{u_i} \cap K_i$, which represents the vertices appearing before the vertex $u_i$ in the true ordering $\pi$.

In both cases, it is important to note that $K_i$ always consists of a contiguous set of vertices (defined by the true ordering) within the clique $K$. Let $s_{i+1}$ and $t_{i+1}$ denote the source and sink vertices, respectively, in the remaining clique $K_{i+1}$. In the first case, where $K_{i+1} = Q_{u_i} \cap K_i$, the vertex $u = u_i$ satisfies $\text{Des}(u) \cap (V(\mathcal{G}) \backslash \text{Des}(s_{i+1})) \cap K = \varnothing$. In the latter case, where $K_{i+1} = P_{u_i} \cap K_i$, the vertex $x = u_i$ satisfies $\text{Des}(t_{i+1}) \cap (V(\mathcal{G}) \backslash \text{Des}(x)) \cap K = \varnothing$. In simpler terms, this means that vertex $u$ (immediate parent) precedes the remaining clique $K_{i+1}$ in the true ordering $\pi$, while vertex $x$ (immediate child) succeeds it within the clique $K$.

Our algorithm terminates when the remaining clique becomes empty, implying that either $u$ is a sink vertex or $u$ and $x$ are consecutive vertices within the clique. In other words, $\text{Des}(u) \cap (V(G) \backslash \text{Des}(x)) \cap K = \varnothing$. Therefore, the solution returned by our algorithm satisfies the conditions of the lemma. The only remaining task is to bound the number of interventions required by our algorithm.

To bound the number of interventions or iterations, we need to analyze the decrease in the size of the clique $K_i$ at each iteration. Recall that $K_i$ consists of a contiguous set of vertices from the original clique $K$, and in each iteration, we randomly select a vertex $u_i$. As discussed earlier, our algorithm either outputs a $1/2$-Meek separator at some intermediate step or performs a randomized binary search to locate the vertices $v_{j^*}$ and $v_{j^*+1}$ (if it exists), satisfying $|A_{v_{j^*}}| \leq |V(\mathcal{G})|/2$ and $|A_{v_{j^*+1}}| > |V(\mathcal{G})|/2$.

Since the parents and children of a vertex $v$ within the clique $K$ are known after intervening on it, the algorithm essentially performs a binary search to find the vertices $v_{j^*}$ and $v_{j^*+1}$. The standard analysis of randomized binary search provides an expected upper bound of $\mathcal{O}(\log |K|)$ iterations.

Combining these insights, we can conclude the proof by establishing that the expected number of iterations is bounded by $\mathcal{O}(\log |K|)$. $\qquad\square$

## D.6 Proof for Theorem 5

**Theorem 5** (Meek separator). *Given an essential graph $\mathcal{E}(\mathcal{G})$ of an unknown DAG $\mathcal{G}$, there exists a randomized procedure* MeekSeparator *(given in Algorithm 1) that runs in polynomial time and*

*adaptively intervenes on a set of atomic interventions $\mathcal{I}$ such that we can find a $1/2$-Meek separator of size at most $2$ and $\mathbb{E}\left[|\mathcal{I}|\right] \leq \mathcal{O}(\log \omega(\mathcal{G}))$,[16] where $\omega(\mathcal{G})$ denotes the size of the largest clique in $\mathcal{G}$.*

*Proof.* The proof of the theorem for a moral DAG follows immediately by combining Lemma 14 and Lemma 10.

In the remainder, we extend our result to encompass general directed acyclic graphs. While an informal argument for general DAGs has been provided at the beginning of Section 4, we now delve into further details to substantiate that argument. Let us recall the definition of $\mathcal{G}^{\mathcal{I}} = \mathcal{G}[E \setminus R(\mathcal{G}, \mathcal{I})]$ and focus on the graph $\mathcal{G}^{\varnothing}$. This subgraph is derived by removing both the v-structure edges and the oriented edges resulting from the application of Meek rules. According to Proposition 17, we establish that $\mathcal{G}^{\varnothing}$ does not introduce any new v-structures and, therefore, is a moral DAG.

To proceed, let $\mathcal{H} \in CC(\mathcal{E}_{\varnothing}(\mathcal{G}))$ be the largest connect component within $\mathcal{E}_{\varnothing}(\mathcal{G})$ and designate $\mathcal{I}$ as a $1/2$-Meek separator of $\mathcal{H}$. Note that $R(\mathcal{H}, \mathcal{I}) \subseteq R(\mathcal{G}^{\varnothing}, \mathcal{I})$ and as highlighted in Proposition 17, we also determine that $R(\mathcal{G}^{\varnothing}, \mathcal{I}) = R(\mathcal{G}, \mathcal{I}) \setminus R(\mathcal{G}, \varnothing)$. Consequently, the connected components within both the intervention essential graphs $\mathcal{E}_{\mathcal{I}}(\mathcal{G})$ and $\mathcal{E}_{\mathcal{I}}(\mathcal{G}^{\varnothing})$ are identical. Since $\mathcal{I}$ is a $1/2$-Meek separator for $\mathcal{G}^{\varnothing}$, it also functions as a half $1/2$-Meek separator for $\mathcal{G}$. $\qquad\square$

# E    Remaining Proofs for Subset Search

Here we present all the proofs for the subset search problem. The proof for the lower bound and the upper bound results are presented in Appendix E.1 and Appendix E.2 respectively.

## E.1    Lower Bound for Subset Search (Lemma 6)

Here we prove our lower bound result for the subset verification problem.

**Lemma 6** (Lower bound). *Let $\mathcal{G} = (V, E)$ be a DAG and $T \subseteq E$ be a subset of target edges, then, $\nu_1(\mathcal{G}, T) \geq \max_{I \subseteq V} \sum_{\mathcal{H} \in CC(\mathcal{E}_I(\mathcal{G}))} \mathbb{1}(E(\mathcal{H}) \cap T \neq \varnothing)$ .*

*Proof.* Consider an intervention set $\mathcal{I}$ and let $CC(\mathcal{E}_{\mathcal{I}}(\mathcal{G}))$ be the set of all connected components in the intervention essential graph $\mathcal{E}_{\mathcal{I}}(\mathcal{G})$. As interventions within each connected components are independent[HB14], we have that,

$$\nu_1(\mathcal{G}, T) \geq \sum_{\mathcal{H} \in CC(\mathcal{E}_{\mathcal{I}}(\mathcal{G}))} \nu_1(\mathcal{H}, T \cap E(\mathcal{H})) .$$

For each $\mathcal{H} \in CC(\mathcal{E}_{\mathcal{I}}(\mathcal{G}))$, where $T \cap E(\mathcal{H})$ is non-empty, it is trivial that $\nu_1(\mathcal{H}, T \cap E(\mathcal{H})) \geq 1$ as we need at least one intervention to orient all the edges in $T \cap E(\mathcal{H})$. Therefore the lower bound follows, which concludes the proof. $\qquad\square$

## E.2    Upper Bound for Subset Search (Theorem 7)

Here we present comprehensive proof for the upper bound result. Although a proof sketch of this result has already been presented in Section 5, we now provide additional details to solidify our argument.

**Theorem 7** (Upper bound). *Let $\mathcal{G} = (V, E)$ be a DAG with $|V| = n$ and $T \subseteq E$ be a subset of target edges. Algorithm 2 that takes essential graph $\mathcal{E}(\mathcal{G})$ and $T$ as input, runs in polynomial time and adaptively intervenes on a set of atomic interventions $\mathcal{I} \subseteq V$ that satisfies, $\mathbb{E}\left[|\mathcal{I}|\right] \leq \mathcal{O}\left(\log n \cdot \log \omega(\mathcal{G})\right) \cdot \nu_1(\mathcal{G}, T)$ and $T \subseteq R(\mathcal{G}, \mathcal{I})$ . Furthermore, in the special case of $T = E$, the solution returned by it satisfies, $\mathbb{E}\left[|\mathcal{I}|\right] \leq \mathcal{O}(\log n) \cdot \nu_1(\mathcal{G})$ and $E \subseteq R(\mathcal{G}, \mathcal{I})$ .*

*Proof.* As discussed in Section 5, the correctness of the Algorithm 2 follows because of the termination condition of the algorithm. Note that, upon termination, all the connected components that encompass the target edges $T$ have a size of $1$. This observation leads to the immediate implication that all the edges belonging to $T$ are oriented.

---

[16]The expectation in the result is over the randomness of the algorithm.

In the remaining part of the proof, we analyze the cost of the algorithm and divide the analysis into two parts: the number of outer loops and the cost per loop.

The bound on the number of outer loops is straightforward. Since the size of connected components containing the target edges decreases at least by a factor of $1/2$ in each iteration, and after $\mathcal{O}(\log n)$ iterations all connected components either consist of a single vertex or do not have any incident target edges $T$, therefore our algorithm terminates in $\mathcal{O}(\log n)$ iterations.

To bound the cost per loop, note that, we invoke the Meek separator only on the connected components $\mathcal{H}$ that have at least one target edge. For each such component $\mathcal{H}$, Lemma 6 establishes that any algorithm must perform at least one intervention on this component. Our Meek separator algorithm performs at most $\mathcal{O}(\log \omega(\mathcal{H})) \in \mathcal{O}(\log \omega(\mathcal{G}))$ interventions for each component $\mathcal{H}$. Hence, in any iteration, we perform at most $O(\log \omega(\mathcal{G}))\nu_1(\mathcal{G}, T)$ interventions. It is important to mention that when $T = E$, we use the lower bound from [SMG$^+$20] which states that any algorithm would require at least $\Omega(\omega(\mathcal{H}))$ interventions to orient all edges within the connected component $\mathcal{H}$, whereas we only perform $O(\log \omega(\mathcal{H}))$ interventions in that iteration. Consequently, in the special case of $T = E$, the cost per iteration is at most $O(\nu_1(\mathcal{G}, E))$.

Combining these analyses, we conclude that our algorithm orients all the edges in $T$ by performing at most $\mathcal{O}(\log n \log \omega(\mathcal{G}))\nu_1(\mathcal{G}, T)$ interventions. In the special case of $T = E$, our total number of interventions is at most $\mathcal{O}(\log n) \cdot \nu_1(\mathcal{G})$. This completes the proof. $\qquad\square$

# F   Remaining Proofs for Causal Mean Matching

Similar to the subset search, we use the Meek separator as a subroutine to provide an approximation algorithm. A crucial step of our algorithm is a source finding algorithm, which given an essential graph $\mathcal{E}(\mathcal{G})$ and a subset of nodes $U \subseteq V(\mathcal{G})$, returns a source node of $U$ by performing at most $\mathcal{O}(\log n \cdot \log \omega(\mathcal{G}))$ number of interventions. Such a source-finding algorithm immediately helps us solve the causal state-matching problem. In the remainder of the section we provide the source finding algorithm and use it solve the causal state matching problem. To understand our source finding algorithm, we need the following lemma.

**Lemma 20.** *Let $\mathcal{G} = (V, E)$ be a DAG, $\mathcal{I} \subseteq V$ be an intervention set and let $S, \mathcal{H} \in \mathcal{E}_{\mathcal{I}}(\mathcal{G}^*)$. Suppose there exists a directed path from $S$ to $\mathcal{H}$, then no directed path exists from $\mathcal{H}$ to $S$. Furthermore, if $s, t \in V$ be such that $t \in \mathtt{Des}(s)$ and they are not in the same connected component, then there exists a directed path that is oriented from the connected component containing $s$ to the connected component containing $t$.*

*Proof.* We are given that there exists a directed path from $S$ to $\mathcal{H}$. As there exists a directed path from $S$ to $\mathcal{H}$, there exist a vertex $s \in V(S)$ and $t \in V(\mathcal{H})$ such that $t \in \mathtt{Des}(s)$.

Suppose for a contradiction assume that there exists a directed path from $\mathcal{H}$ to $S$. Let $u$ and $v$ be the endpoint of this directed path in $\mathcal{H}$ and $S$ respectively. By Proposition 17, we know that all the recovered parents for the vertices within the same connected component are the same; therefore we have that, $\mathtt{Pa}_{u,\mathcal{I}}(\mathcal{G}) = \mathtt{Pa}_{t,\mathcal{I}}(\mathcal{G})$ and $\mathtt{Pa}_{s,\mathcal{I}}(\mathcal{G}) = \mathtt{Pa}_{v,\mathcal{I}}(\mathcal{G})$. Let $t'$ be the parent of $t$ on the directed path that connects the vertices $s$ and $t$. Note that $t' \in \mathtt{Des}[s]$ and $t'$ could be $s$ or some other vertex. As $s$ and $t$ belong to different connected components $t' \to t$ is oriented. As $\mathtt{Pa}_{u,\mathcal{I}}(\mathcal{G}) = \mathtt{Pa}_{t,\mathcal{I}}(\mathcal{G})$, we have that $t' \in \mathtt{Pa}_{u,\mathcal{I}}(\mathcal{G})$, therefore $u \in \mathtt{Des}(t')$ which further implies $u \in \mathtt{Des}(s)$. As $v \in \mathtt{Des}(u)$ and as they are in different connected components, a similar argument as above implies that $s \in \mathtt{Des}(u)$, which is a contradiction. Therefore $\mathcal{H}$ to $S$ directed path does not exist. The analysis conducted above verifies the first claim of the lemma. In the subsequent portion, we focus on establishing the second part.

Consider vertices $s$ and $t$ such that $t \in \mathtt{Des}(s)$. If both $s$ and $t$ belong to the same connected component in the interventional essential graph $\mathcal{E}_{\mathcal{I}}(\mathcal{G})$, the lemma statement holds trivially. Henceforth, we assume that $s$ and $t$ belong to distinct connected components, denoted as $S$ and $T$ respectively.

Since $t \in \mathtt{Des}(s)$, there exists a directed path from vertex $s$ to vertex $t$ in the ground truth DAG $\mathcal{G}$. Let us denote $Q$ as the shortest directed path from $s$ to $t$ in $\mathcal{G}$. To form path $P$, we remove the edges within the connected components $S$ and $T$ from $Q$ and retain only the edges that connect $S$ and $T$. Consequently, $P$ represents this modified portion of path $Q$. Furthermore, let $v$ and $w$ be the

respective endpoints of path $P$ in $S$ and $T$. As $S$ and $T$ are two different connected components, we have that some of the edges in $P$ are oriented.

Suppose that all edges in $P$ are oriented. In this case, we have already found a directed path from $S$ to $T$ in $\mathcal{E}_{\mathcal{I}}(\mathcal{G})$, and our objective is achieved. Hence, we focus on the scenario where some edges in $P$ are unoriented. Let $P : v = v_0 \to v_1 \to \cdots \to v_\ell \to w$ denote the vertices along the path $P$, and let $v_j \sim v_{j+1}$ represent the first unoriented edge encountered.

Since $v_1$ does not belong to the connected component $S$, we have that $v \to v_1$ is oriented and we have $v_j \neq v$. Now, consider the situation where $v_{j-1}$ and $v_{j+1}$ are not adjacent. In such a case, the Meek rule R1 would orient the edge $v_j \to v_{j+1}$. However, since $v_j \to v_{j+1}$ remains unoriented, it implies that the edge $v_{j-1} \to v_{j+1}$ must exist. However, this contradicts the fact that $P$ is the shortest directed path. Therefore, it must be the case that $P$ is either entirely oriented or entirely unoriented. As some edges in $P$ are oriented, we conclude that path $P$ is completely oriented, thereby establishing the presence of a directed path from $S$ to $T$ in the interventional essential graph $\mathcal{E}_{\mathcal{I}}(\mathcal{G})$, which concludes the proof. □

### F.1 Proof for Lemma 8

We now provide the analysis of the source-finding algorithm. The guarantees of this algorithm are summarized in the lemma that follows.

**Lemma 8** (Source finding). *Let $\mathcal{G} = (V, E)$ be a DAG and $U \subseteq V$ be a subset of vertices. Algorithm 3 that takes essential graph $\mathcal{E}(\mathcal{G})$ and $U$ as input, runs in polynomial time and adaptively intervenes on a set of atomic interventions $\mathcal{I} \subset V$, identifies a source vertex of the induced subgraph $\mathcal{G}[U]$ with $\mathbb{E}[|\mathcal{I}|] \leq \mathcal{O}\big(\log n \cdot \log \omega(\mathcal{G})\big).$*

*Proof.* As $\mathcal{J}_i$ is a $1/2$-Meek separators, we immediately have that $|V(\mathcal{H}_{i+1})| \leq |V(\mathcal{H}_i)|/2$. Therefore our algorithm terminates in at most $\mathcal{O}(\log n)$ iterations. By Theorem 5, note that, in each iteration, we make at most $O(\log \omega(\mathcal{H}_i)) \in \mathcal{O}(\log \omega(\mathcal{G}^*))$ number of iterations. Therefore, the total number of interventions are at most $\mathcal{O}(\log n \cdot \log \omega(\mathcal{G}^*))$. It remains to show that we find the source node of $U$.

As in each iteration we recurse on connected component $\mathcal{H}_i \in C_i$ that has no incoming edge from any other component $\mathcal{H} \in C_i$. From Lemma 20, we know that $\mathcal{H}_i$ contains one of the source nodes of $U$. Therefore our algorithm always recurses on a connected component containing a source node until the algorithm finds it. □

---

**Algorithm 4** CausalMeanMatch($\mathcal{E}(\mathcal{G}), P, \mu^*$)

---

1: **Input**: Essential graph $\mathcal{E}(\mathcal{G})$ of a DAG $\mathcal{G}$, observational distribution of $V$, and desired mean $\mu^*$.
2: **Output**: Atomic intervention set $\mathcal{I}$.
3: Initialize $\mathcal{I}^* = \varnothing$ and $\mathcal{I} = \{\varnothing\}$.
4: **while** $\mathbb{E}_{P^{\mathcal{I}^*}}(V) \neq \mu^*$
5:      Let $T = \{i | i \in [p], \mathbb{E}_{P^{\mathcal{I}^*}}(V_i) \neq \mu_i^*\}$.
6:      Let $\mathcal{G}$ be the subgraph of $\mathcal{E}_{\mathcal{I}}(\mathcal{G})$ induced by $T$.
7:      Let $U_T$ be the identified source nodes in $T$.
8:      **while** $U_T = \varnothing$
9:          Let $\mathcal{C}$ be a chain component of $\mathcal{G}$ with no incoming edges.
10:          Perform interventions by FindSource($\mathcal{E}(\mathcal{G}), S$) and append interventions to $\mathcal{I}$.
11:          Update $\mathcal{G}$ and $U_T$ as the outer loop.
12:      Set $a_i = \mu_i^* - \mathbb{E}_{P^{\mathcal{I}^*}}(V_i)$ for $i$ in $U_T$.
13:      Include the atomic interventions with perturbation targets $U_T$ and shift values $\{a_i\}_{i \in U_T}$ respectively in $\mathcal{I}^*$ and perform $\mathcal{I}^*$.
14: **return** $\mathcal{I}^*$

---

### F.2 Proof for Theorem 9

Given such a source-finding algorithm, we can use it to solve the mean matching problem. We summarize this result below.

**Theorem 9** (Causal mean matching). *Let $\mathcal{G}$ be a DAG and $\mathcal{I}^*$ be the unique solution to the causal mean matching problem with desired mean $\mu^*$. Algorithm 4 that takes $\mathcal{E}(\mathcal{G})$ and $\mu^*$ as input, runs in polynomial time and adaptively intervenes on set $\mathcal{I} \subseteq V$, identifies $\mathcal{I}^*$ with $\mathbb{E}[|\mathcal{I}|] \le \mathcal{O}\big( \log n \cdot \log \omega(\mathcal{G}) \big) \cdot |\mathcal{I}^*|$.*

*Proof.* Note the outer loop in Algorithm 4 takes at most $|\mathcal{I}^*|$ round. In each of this round, the inner loop is ended with at most $\mathcal{O}(\log n \log \omega(\mathcal{G}^*))$ interventions in expectation, as proven by Lemma 8. Therefore, in expectation, the number of interventions performed is upper bounded by $\mathcal{O}(\log n \log \omega(\mathcal{G}^*))|\mathcal{I}^*|$. $\qquad\square$

## G   Details of Numerical Experiments

We implemented our methods using the NetworkX package [HSSC08] and the CausalDAG package https://github.com/uhlerlab/causaldag. All code is written in Python and run on CPU. The source code of our implementation can be found at https://github.com/uhlerlab/meek_sep.

### G.1   Subset Search

**Problem Generation:** We consider the $r$-hop model in [CS23]. In this model, an Erdös-Rényi graph [ER60] with edge density $0.001$ on $n$ nodes is first generated. Then a random tree on these $n$ nodes is generated. The final DAG is obtained by (1) combining the edge sets using a fixed topological order, where $u \to v$ if it is in the combined edge sets and $u$ has a smaller vertex label than $v$, and (2) removing v-structures by connecting $u \to w$ where $u \to v \leftarrow w$ and $u$ has a smaller vertex label than $w$. Then the subset of edges is selected to be the edges within $r$-neighborhood of a randomly picked vertex.

**Multiple Runs:** For each dot presented in the results, we run each method on 20 different instances using the generation method described above. The average and standard deviation across instances are reported in the results.

Figure 8 shows similar results as Figure 5a on the 3-hop model, where we vary the number of hops $r \in \{1, 2, 4, 5\}$ on DAGs with different sizes. Figure 9 shows the trend of varying number of hops on DAGs with different sizes. We observe our method MeekSep and MeekSep-1 to consistently outperform existing baselines.

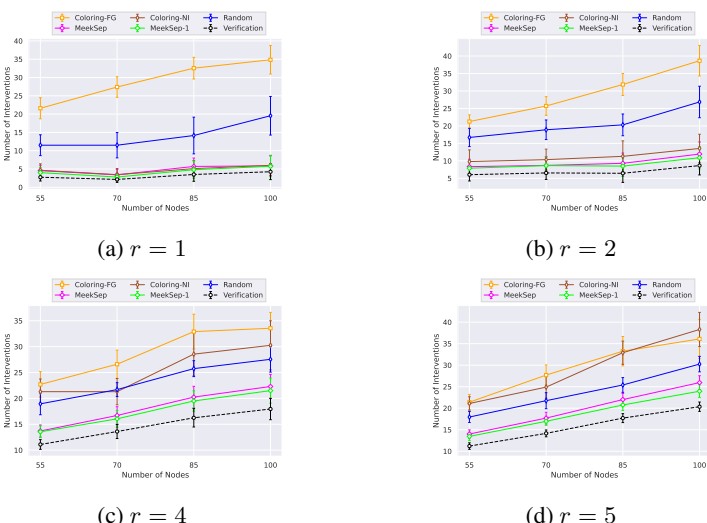

(a) $r = 1$  (b) $r = 2$

(c) $r = 4$  (d) $r = 5$

Figure 8: Meek separator for subset search on $r$-hop model. Each dot is averaged across 20 DAGs, where the error bar shows $0.5$ standard deviation.

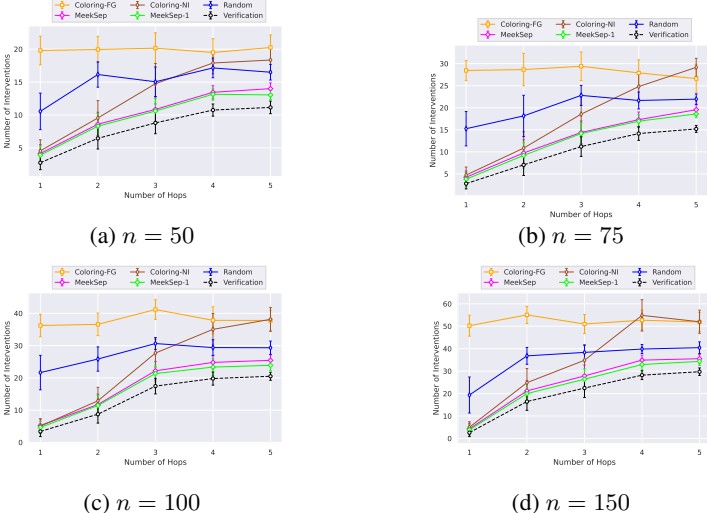

(a) $n = 50$

(b) $n = 75$

(c) $n = 100$

(d) $n = 150$

Figure 9: Meek separator for subset search on $r$-hop model. Each dot is averaged across 20 DAGs, where the error bar shows $0.5$ standard deviation.

## G.2 Causal Mean Matching

**Problem Generation:** We consider three random graph models: Erdös-Rényi graphs, Barabási–Albert graphs [AB02], and random tree graphs. The edge density in Erdös-Rényi graphs is $0.2$ where the number of edges to attach from a new node to existing nodes in Barabási–Albert graphs is set to 2. The intervention targets of $\mathcal{I}^*$ is a random subset of $n$ vertex in the DAG.

**Multiple Runs:** For each dot presented in the result, we run each method on 10 different instances using the generation method described above. The average and standard deviation across instances are reported in the results.

Figure 5b shows the result on Erdös-Rényi graphs, where we vary the number of targets in $\mathcal{I}^*$ on DAGs with 50 nodes. Figure 10a and Figure 10b show similar results on random tree graphs and Barabási–Albert graphs. In Figure 10c, we consider Erdös-Rényi graphs where $|\mathcal{I}^*|$ is set to 25. This result shows the trend of varying number of nodes. We observe that our method is empirically competitive with the state-of-the-art method `CliqueTree` across all cases.

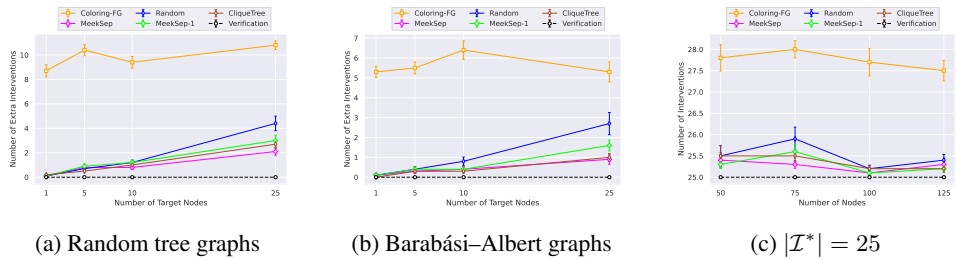

(a) Random tree graphs

(b) Barabási–Albert graphs

(c) $|\mathcal{I}^*| = 25$

Figure 10: Meek separator for causal matching. Each dot is averaged across 10 DAGs, where the error bar shows $0.2$ standard deviation

