# OpenReview forum: "Meek Separators and Their Applications in Targeted Causal Discovery"
_NeurIPS.cc/2023/Conference — NeurIPS 2023 poster_

### Official Review · Reviewer_wSxh · 2023-06-27

**Soundness:** 3 good
**Presentation:** 3 good
**Contribution:** 3 good
**Rating:** 6
**Confidence:** 2

**Summary:**

The paper focuses on applications of causal discovery, in which it is not necessary to learn the full graph. Instead, the authors propose to recover what they call the Meek separator---which consists of a set of vertices that decomposes the unoriented edges into smaller connected components when intervened on. Further, they propose two randomized algorithms (for subset search and causal mean matching) for which they prove logarithmic approximation guarantees.

**Strengths:**

- The authors provide strong theoretical guarantees (logarithmic approximation expectation) for both of their proposed randomized algorithms.
- On synthetic data, the proposed algorithms perform well compared to the baselines.
- The code for the experiments is provided.


**Weaknesses:**

- The paper is very dense and hard to follow. Especially, I would appreciate it if more effort was put into motivating the main concepts---i.e., why do we want to learn the Meek separator, and why is the definition reasonable? Figure 1 is supposed to provide an example, but very little intuition is provided to guide the reader through the proposed concepts.
- Although the problem is motivated by learning partial information from gene expression networks and referring to [FMT+21], no real-world example was studied.

Minor feedback:
- In the introduction, it is stated that only the Markov equivalence class can be learned from observational data. This is only partially true. The Markov equivalence class can be learned assuming, e.g., faithfulness and the causal Markov property, but with different assumptions, e.g., assumptions w.r.t. the SCM (linear non-Gaussian additive noise), the DAG is identifiable.
- Line 84: The $\sim$ relation is used to indicate if two vertices are “connected”. I think the notion of “adjacent” is more common since connected could also mean that the path between the vertices has length $> 1$.


**Questions:**

- It would be great if the authors could provide more intuition as to why the Meek separator is a good set of vertices to learn. For example, if we are interested in the neighborhood of a node, we could also learn the Markov blanket---which advantages does the Meek separator bring with it?
- In Figure 1, why can we ignore all directed edges? The intervention on 2 should only delete the edge $1 \to 2$.


**Limitations:**

- No real-world examples have been provided.
- To me, it is unclear what exactly the benefits of the proposed Meek separator are, and how it relates to, e.g., Markov blankets.

---

> ### Author Rebuttal · Authors · 2023-08-09
>
> We appreciate that the reviewer found our theoretical results to be strong. We would like to address some of the reviewer’s comments below:
>
> > **”motivating the main concepts---i.e., why do we want to learn the Meek separator, and why is the definition reasonable? Figure 1... guide the reader through the proposed concepts.”**
>
> We are motivated to study the Meek separator due to its ability to break down *the remaining unoriented edges* of the DAG into more manageable connected components upon interventions. As emphasized in both the abstract and introduction, the Meek separator discovery algorithm holds promise for enabling divide-and-conquer strategies in solving diverse targeted causal discovery challenges. Specifically, our research showcases its application in crafting approximation algorithms for problems like subset search and mean matching.
>
> Furthermore, this definition is in contrast with the traditional *graph separator*, which decomposes the *full graph* (instead of *the remaining unoriented edges*) into smaller connected components (Definition 1). Note that the graph separator does not make use of the information of interventions. Since we are able to learn some edge orientations after interventions, the set of *unoriented edges* can be much smaller than the set of edges in the *full graph*. Therefore the size of a Meek separator can be much smaller than the size of a graph separator. Figure 1 aims to illustrate this.
>
> In the *PDF attached to the general response,* we provided an updated version of Figure 1 (which will be used in the revised manuscript). In particular, we added detailed explanations of the Meek separator and a separate panel illustrating the graph separator. Figure 1d shows the 1/2-graph separator which contains 2 vertices. Since each pair of vertices is adjacent, removing 1 vertex will leave a connected component of size 4-1=3>2. Therefore any 1/2-graph separator must contain at least 2 vertices. However, Figure 1b shows that there exists a Meek separator that contains only 1 vertex.
>
>
> > **”no real-world example was studied.”**
>
> We thank the reviewer for this comment. In our experiments, we used synthetic simulations as a means to demonstrate how our algorithm can be useful in real-world applications. These simulations are idealized models for real-world experiments. For example, the task of mean matching can be used as an abstraction of a cell reprogramming experiment, where the shift interventions can be used to model gene over-expression or knockdowns (as discussed in the introduction).
>
> While we acknowledge the importance of real-world experiments, one major challenge lies in the nature of the adaptive policy algorithm, where access to real-world sequential data is not readily available. Similar constraints are present in previous papers in this line of work (e.g., [1-4]), where the evaluations are often based on synthetic data. Moreover, the implementation of these algorithms in real-world applications necessitates close collaboration with individuals engaged in experimentation. Going forward, we are actively considering collaborations with experimentalists or the utilization of semi-synthetic data generated from real experiments. However, much work remains to be done to benchmark these real datasets and thus we consider it out of scope for this work.
>
> > **”In the introduction, it is stated that only the Markov equivalence class can be learned from observational data. This is only partially true. The Markov equivalence class can be learned assuming, e.g., faithfulness and the causal Markov property, but with different assumptions... ”**
>
> We thank the reviewer for pointing this out. In the introduction, we stated that a DAG is *generally* only identifiable up to its MEC with observational data. But we will make this clearer by adding a statement that this result holds in nonparametric SCMs while additional identifiability can be achieved by considering parametric SCMs (e.g., linear SCMs with non-Gaussian additive noises).
>
> > **”Line 84: The ∼ relation is used to indicate if two vertices are “connected”. I think the notion of “adjacent” is more common since connected could also mean that the path between the vertices has length >1.”**
>
> Thank you for this comment. We will revise “connected” to “adjacent” accordingly.
>
> > **”provide more intuition as to why the Meek separator is a good set of vertices to learn... if we are interested in the neighborhood of a node, we could also learn the Markov blanket---which advantages does the Meek separator bring with it?”**
>
> See above for the intuition of why we want to learn the Meek separator and its benefits. Regarding Markov blankets, we do not see a way to make direct comparisons. First, Markov blankets can be learned from the observational data, while our work aims to learn a pre-specified subset of edges from interventional data using the least number of interventions. Second, an identified Markov blanket does not distinguish causes from effects (i.e., edge orientations) [5], while we aim to learn edge orientations in subset search. More importantly, our goal lies not only in learning the neighborhood of a node, but learning an arbitrary set of edges for the subset search problem or a matching intervention for the mean matching problem.
>
> > **”In Figure 1, why can we ignore all directed edges? The intervention on 2 should only delete the edge 1$\rightarrow$ 2.”**
>
> Intervening on a vertex allows us to identify all edges adjacent to it and possibly additional edges given by the Meek rules (Appendix A). See lines 125-127 for details. Therefore in Figure 1b, when we intervene on vertex 2, it allows us to identify the edges $1\rightarrow 2, 2\rightarrow 3, 2\rightarrow 4$ and $1\rightarrow 3, 1\rightarrow 4$ by Meek rules. Then in Figure 1c, we show the connected components of the graph after removing all the oriented edges.
>
> ---
> All references in this response can be found in the general response.

---

> > ### Comment · Reviewer_wSxh · 2023-08-15
> >
> > Thank you for the clarifications! I have read the other reviews and the rebuttal and do not see any major concerns regarding the paper. I hope that the authors incorporate the feedback from the reviewers and aim to improve the readability of the paper to make it more accessible to a broader audience. I adjusted my score to "weak accept".

---

> > > ### Author Response · Authors · 2023-08-15
> > > **Thank you for the reply and discussions!**
> > >
> > > Thank you for the reply and discussions! We will incorporate the feedback into our revision. Let us know if there are any additional questions!

---

### Official Review · Reviewer_2LUY · 2023-07-04

**Soundness:** 3 good
**Presentation:** 3 good
**Contribution:** 3 good
**Rating:** 7
**Confidence:** 3

**Summary:**

This paper studies the problem of learning causal structure by learning a minimal intervention set, which is formalized as the Meek separator. The authors show that the Meek separator can orient the maximum number of edges with the minimum intervention set while limiting the size of the remaining undirected connected components to $\alpha |V|$. Furthermore, the authors provide a logarithmic-time algorithm to determine the variables of the Meek separator based on a binary search on the essential graph.  Also, the authors show that the problems of subset search and causal matching can be well addressed by the Meek separator.


**Strengths:**

1. This paper provides a Meek separator and algorithm that only require logarithmic time complexity, which addresses an important problem of causal discovery.


2. The proposed Meek separator can be flexibly applied to subset search and causal matching. The experimental results verify the effectiveness of the proposed algorithms.

3. The proposed theorem looks sound.

**Weaknesses:**

1. Section 2 is poor readability due to many notions and symbols.  I suggest the definition part and related work can be divided into two subsections.

2. I think some examples should be provided for illustrating significant graphical concepts, such as moral graph and essential graph.


3. The result of Lemma 10 is interesting but is not easy to understand. Can you provide an intuitive explanation for it?

**Questions:**


Can your method be directly applied to learn the entire causal graph？Will there be any new challenges?


**Limitations:**

Refer to Weaknesses

---

> ### Author Rebuttal · Authors · 2023-08-09
>
> Thank you for appreciating the problem we studied, and for your recognition of our result! We’ve added illustrative examples as per your suggestion and we’d like to address your concerns below:
>
> > **”Section 2 is poor readability due to many notions and symbols. I suggest the definition part and related work can be divided into two subsections.”**
>
> Thank you for this feedback. When we originally wrote the definitions in Section 2, it seemed to be a natural place to refer to related work, as these concepts were first defined in those works. However, to improve readability, we will divide this section into subsections describing:
>
> - basic graphical concepts (paragraph 1, lines 81-97),
> - graphical concepts relating to DAGs and causal models (paragraph 2, lines 98-116), and
> - concepts involving interventions (paragraphs 3-6, lines 117-152).
>
> In addition, we will also add a few illustrative examples describing some key definitions in Appendix B (see our response to the other comment regarding examples below).
>
> > **”I think some examples should be provided for illustrating significant graphical concepts, such as moral graph and essential graph.”**
>
> Thank you for this suggestion! In the *PDF attached to the general response*, we added a few examples illustrating the key graphical concepts. In particular,
>
> - *Figure 3* illustrates a moral graph versus a graph that is not moral. We will add this example to Appendix B and refer to it when moral DAGs are introduced in Section 2.
> - *Figure 4* illustrates the essential graph, $\mathcal{I}$-essential graph, and connected components. We will add this example to Appendix B and refer to it when they are defined in Section 2.
> - In addition, *Figure 1* now includes a separate panel illustrating the traditional graph separator and detailed explanations of our defined Meek separator. We will replace Figure 1 in the revised manuscript with this.
>
> > **”The result of Lemma 10 is interesting but is not easy to understand. Can you provide an intuitive explanation for it?”**
>
> Thank you for this suggestion! Intuitively, we can explain the result of Lemma 10 below.
>
> Lemma 10 establishes the existence of a subset of at most two vertices that form a Meek separator. It also shows that this subset will satisfy several nice properties. These properties empower us (as depicted in Algorithm 1) to discover such subsets using a binary search method within the vertices of the provided 1/2-clique separator.
>
> To elaborate, the statement of Lemma 10 is three-fold. Firstly, it states that there is a vertex u that is almost central to the graph (i.e., fulfills the **constraint** “$|A_u| ≤ |V(\mathcal{G})|/2$ and $|A_v | > |V (\mathcal{G})|/2$ for all $v ∈ Des(u) \cap K$”). Then, it states that any vertex that fulfills this constraint will satisfy one of the two **conditions** (i.e., “ 1). either $u$ is a sink vertex …  2)…..”). Thirdly, it states that the $u$ (and potentially $x$) stated in the conditions corresponds to Meek separators. Consequently, by finding a vertex that fulfills the constraint, we find the Meek separators by the conditions it satisfies.
>
> See also the general response for an intuitive explanation of its proof.
>
>
> > **”Can your method be directly applied to learn the entire causal graph? Will there be any new challenges?”**
>
> Yes. Our algorithm for subset search can be directly applied to learn the entire causal graph, by specifying the target edges to be the set of all edges. In this case, we get a $\mathcal{O}(\log n)$ approximation for recovering the entire DAG, which matches the current best approximation ratio in [4]. This discussion can be found in lines 208-209.
>
> ---
> The references in this response can be found in the general response.

---

> ### Comment · Reviewer_2LUY · 2023-08-10
> **I appreciate the responses from the authors**
>
> I appreciate the responses from the authors. Most of my concerns are addressed, so I will change my score to "Accept". I suggest the authors incorporate the responses to the final version of the paper if accepted.

---

> > ### Author Response · Authors · 2023-08-11
> > **We greatly appreciate your prompt response and the valuable feedback.**
> >
> > We greatly appreciate your prompt response and the valuable feedback. We will certainly integrate these suggested changes into our revision. Additionally, please feel free to reach out if there are any further questions.
> >
> > On an additional note, we just want to highlight that we don't see any change in the score from "weak accept" to "accept." Is this something that was anticipated? If so, our sincere apologies for bringing up this matter.

---

> > > ### Comment · Reviewer_2LUY · 2023-08-12
> > >
> > > Thanks for the reminder, I have raised my score.

---

### Official Review · Reviewer_Zms6 · 2023-07-07

**Soundness:** 3 good
**Presentation:** 3 good
**Contribution:** 3 good
**Rating:** 7
**Confidence:** 2

**Summary:**

This work explores the problem of learning the local causal structure from intervention data. Specifically, the authors introduce a novel separator called the $\alpha$-Meek Separator. Unlike traditional $\alpha$-Separators, their separator imposes bounds on the sizes of connected components in a subgraph. The authors also apply the Meek separator to address two typical tasks: subset search and causal matching. Furthermore, they validate the effectiveness of their methods through experiments conducted on synthetic datasets.

**Strengths:**

The paper tackles a very challenging problem.

The meek separator introduced in this paper is novel and interesting.

The applications of the Meek Separator in subset search and causal matching are both significant.

The analysis in this paper is presented in a logical manner.

**Weaknesses:**

Regarding readability: The majority of the article consists of descriptive statements that provide definitions and conclusions but lack simple examples for illustration. In order to help more readers understand and learn, it is recommended to provide some basic examples for key definitions and algorithms.

**Questions:**

See above.

**Limitations:**

All theoretical results are based on the assumption that all variables are observable, meaning that there are no hidden variables.

---

> ### Author Rebuttal · Authors · 2023-08-09
>
>
> Thank you for the encouraging comments! We appreciate that you think our proposal is novel and significant. We’ve added illustrative examples for better readability and we’d like to address your comments here:
>
> > **”Regarding readability: The majority of the article consists of descriptive statements that provide definitions and conclusions but lack simple examples for illustration. In order to help more readers understand and learn, it is recommended to provide some basic examples for key definitions and algorithms.”**
>
> Thank you for this suggestion. In the *PDF attached to the general response*, we added a few examples illustrating the key definitions and algorithms. In particular,
>
> - *Figure 1* now includes detailed explanations of our defined Meek separator and a separate panel illustrating the traditional graph separator. We will replace Figure 1 in the revised manuscript with this.
> - We added *Figure 2* to show how Algorithm 1 finds the Meek separator in a 4-vertex DAG example. In this example, we walk through the iterations of Algorithm 1 while specifying the realizations of the selected $u_i$ in line 4 of Algorithm 1. We will add this example to section 4 of the revised manuscript.
> - *Figure 3*  and *Figure 4* illustrate key graphical concepts that were used, including moral graphs, essential graphs, and connected components. We will add these examples to Appendix B and refer to these figures when they are introduced in Section 2.
>
> > **”All theoretical results are based on the assumption that all variables are observable, meaning that there are no hidden variables.”**
>
> We appreciate this comment regarding our assumptions of no hidden variables. We agree that it would be very interesting to explore settings with latent confounders. We consider this as an important aspect to extend this line of work that often assumes causal sufficiency, and we hope to address this further in future works.

---

> > ### Comment · Reviewer_Zms6 · 2023-08-12
> > **Thanks for your response**
> >
> > Thanks for your response, after reading the response and other reviewers' comments, I will keep my score.

---

> > > ### Author Response · Authors · 2023-08-14
> > > **Thank you for the discussion**
> > >
> > > Thank you for the discussion! We are happy to address further comments if there are any.

---

### Official Review · Reviewer_BZ2Z · 2023-07-26

**Soundness:** 3 good
**Presentation:** 3 good
**Contribution:** 3 good
**Rating:** 6
**Confidence:** 3

**Summary:**

* The paper provides an algorithm for finding a subset of vertices in a causal graph that, when intervened, can turn undirected edge into smaller connected components for learning a part of the causal graph.
* The proposed algorithm comes with first known average-case provable guarantees for two applications:  subset search and causal matching.

**Strengths:**

* The authors propose a randomized algorithm that finds an intervention set of small size that, when intervened, decomposes the remaining undirected edges into connected components of smaller sizes.
* The authors demonstrate the utility of the proposed targeted causal discovery algorithm in the problems of subset search and causal mean matching with analysis to show exponential improvements upon existing results in those domains.


**Weaknesses:**

* It would be advisable to add examples to show how the algorithm finds a Meek separator.
* The paper concerns problems motivated by practical scenarios, but the paper lacks real-world experiments to demonstrate the utility of the proposed algorithm.
* The proofs of the theorems and lemma often omit details. Please see the questions below.


**Questions:**

* Suggestions:
   * It would be good to use an example that both illustrates the idea of Meek separator and the limitations of the graph separator in Figure 1. Also, use the same example to show why any alpha-graph separator must contain at least $(1-\alpha) |V|$ vertices.
   * Add the description of w(G) in the Lemma 2 like theorem 5.
   * Lemma 2 should cite Theorem 1 and Theorem 3 in the reference.
   * Defining what “joins” means in Definition 1 would be good.
   * In line 272, “the set $u,x$” should have curly brackets around $u, x$. Same for line 277.
   * In line 536 of the supplement, $H$ should be $\mathcal{H}$
   * In line 575, $V(\mathcal{H}) = V(\mathcal{G}) \setminus Des[u]$ should be $V(\mathcal{H}) \subseteq V(\mathcal{G}) \setminus Des[u]$.

* Questions:
  * The intervention set $\mathcal{I}$ is defined as a set of sets right above definition 3, shouldn’t Figure 1b write $\mathcal{I} = \{\{ 2\} \} $?
  * In line 267, the authors use $V(\mathcal{H}) \cap V(K) = \emptyset$, in line 270, the authors use $V(\mathcal{H}) \cap K = \emptyset$, what is the difference between $K$ and $V(K)$?
  * Is it possible to have different Meek separators where one is preferable to the other per iteration?
  * Will the performance of the algorithms be different for soft interventions and hard interventions in the experiment?
  * Regarding Lemma 11, how can you conclude intervening on $v$ orients the edge $(c,d)$ without proving $v \in Des[w] \cap Anc[d]$ for some $w \in Anc[c]$ for any $v$ as described by Lemma 18?
  * In the proof of Lemma 10, in line 573, it says "It is important to note that u fulfills the conditions specified in the lemma.", can you specify on which condition specifically? Also, in line 574, why does it need to suppose $u$ is a sink vertex of $K$ when $u$ is defined as the last vertex in terms of true ordering within $K$?





**Limitations:**

yes.

---

> ### Author Rebuttal · Authors · 2023-08-09
>
>
> Thank you so much for your detailed review, and for acknowledging our randomized algorithm! We would like to address your points below:
>
> > **”add examples to show how the algorithm finds a Meek separator.”**
>
> We thank the reviewer for this suggestion. The added examples and detailed explanations can be found in the general response.
>
> > **”lacks real-world experiments to demonstrate the utility of the proposed algorithm.”**
>
> In our paper, we used synthetic simulations as a means to demonstrate how our algorithm can be useful in real-world applications. These simulations serve as idealized models for real-world experiments. For example, the task of mean matching can be used as an abstraction of a cell reprogramming experiment, where the shift interventions can be used to model gene over-expression or knockdowns (as discussed in the introduction).
>
> While we acknowledge the importance of real-world experiments, one major challenge lies in the nature of the adaptive policy algorithm, where access to real-world sequential data is not readily available. Similar constraints are present in previous papers in this line of work (e.g., [1-4]), where the evaluations are often based on synthetic data. Moreover, the implementation of these algorithms in real-world applications necessitates close collaboration with individuals engaged in experimentation. Going forward, we are actively considering collaborations with experimentalists or the utilization of semi-synthetic data generated from real experiments. However, much work remains to be done to benchmark these real datasets and thus we consider it out of scope for this work.
>
>
> > **”an example that both illustrates the idea of Meek separator and the limitations of the graph separator in Figure 1... why any $\alpha$-graph separator must contain at least $(1-\alpha) |V|$ vertices.”**
>
> We thank the reviewer for this suggestion. In *Figure 1* in the *PDF attached to the general response* (which will replace Figure 1 in the revised manuscript), we added detailed explanations of the Meek separator as well as a separate panel illustrating a graph separator. Figure 1d shows the 1/2-graph separator which contains 2 vertices. Since each pair of vertices is adjacent, removing 1 vertex will leave a connected component of size 4-1=3>2. Therefore any 1/2-graph separator must contain at least 2 vertices.
>
> This example shows a 4-vertex fully connected DAG and a 1/2-graph separator. It can be extended to fully connected DAGs with $|V|$ vertices and $\alpha$-graph separators. Since every pair of vertices in the clique is adjacent, removing $<(1-\alpha)|V|$ vertices will leave a connected component of size $>\alpha |V|$. Thus any $\alpha$-graph separator in this case must contain at least $(1-\alpha)|V|$ vertices. This discussion is provided in lines 162-164.
>
> > **”Add the description of w(G) in the Lemma 2”**
>
> In Lemma 2, we stated $\omega(\mathcal{G})$ as "vertices in its largest clique”. However, we will make this clearer by separating this description into its own sentence.
>
> Thank you for the other suggestions as well:
>
> > **”Lemma 2 should  …  should be** $V(\mathcal{H})\subset V(\mathcal{G})\subset Des[u]$**.”**
>
> We will add detailed pointers to Theorem 1,3 in the reference in Lemma 3 and change lines 272, 277, 536, and 575 as suggested.
>
> > **”shouldn’t Figure 1b write $\mathcal{I}=${{2}}?”**
>
> Thank you for pointing this out. $\mathcal{I}$ is defined to be the set of sets and therefore it should be $\mathcal{I}$={{2}}.
>
> > **“In line 267, the authors use $V(\mathcal{H})\cap V(K)=\varnothing$, in line 270, the authors use $V(\mathcal{H})\cap K=\varnothing$, what is the difference between $K$ and $V(K)$?”**
>
> We use $K$ to denote the graph and $V(K)$ to denote its vertices. Therefore, it should be $V(\mathcal{H})\cap V(K)=\varnothing$ in line 270. We tried to change all the vertex sets referred to as $V(\cdot)$; however, we missed some. In the revised version, we will change the remaining ones.
>
> > **”different Meek separators where one is preferable to the other?”**
>
> For the current set of applications, the Meek separators are only used to break down a larger graph into smaller, manageable subgraphs. In this context, any Meek separator that can be efficiently computed with minimal interventions provides the same guarantee. Therefore, for our current use cases, we do not find a significant difference between different Meek separators, and they all seem to be equally effective in solving the problems.
>
> > **”performance of the algorithms different for soft interventions and hard interventions in the experiment?”**
>
> No, the performance of the algorithms is not different for soft interventions and hard interventions in our experiment. The property of the interventions that we utilized is to identify additional edges. For both soft and hard interventions, the identifiability of the interventions remains the same, where we can learn the orientations of any edge cut by $I$ and $V\setminus I$ (here $I$ denotes the vertices being intervened and $V$ denotes all the vertices in the graph) and possibly additional edges given by the Meek rules (Appendix A). This discussion can be found in lines 125-127.
>
> > **”Regarding Lemma 11, ..., described by Lemma 18?”**
>
> Thank you for pointing this out. Lemma 18 establishes that the edge $(c,d)$ is orientable through intervention on any vertex within the set $Des[w] \cap Anc[d]$ for some fixed $w \in Anc[c]$. As $w \in Anc[c]$, we have that $Des[c] \subseteq Des[w]$, and therefore $Des[c] \cap Anc[d] \subseteq Des[w] \cap Anc[d]$; consequently as $v \in Des[c] \cap Anc[d]$, we immediately get that $v$ orients the edge $(c,d)$. We will incorporate these lines into our proof to enhance its readability.
>
> > **”In the proof of Lemma 10...”**
>
> We apologize for the confusion! Due to character limits, we gave detailed clarifications in the general response.
>
> ---
> All references in this response can be found in the general response.

---

> > ### Comment · Reviewer_BZ2Z · 2023-08-11
> > **Follow-up questions**
> >
> > I appreciate the authors' response. Does the $\mathcal{I}$ in Figure 2 caption in the newly attached pdf file mean a different thing? Why is it not $\mathcal{I} =$ {{$2$}}?

---

> > > ### Author Response · Authors · 2023-08-11
> > > **Response to follow-up questions**
> > >
> > > Thank you for the quick response! We are happy to clarify the follow-up question:
> > > - $\mathcal{I}$ in Figure 2 caption refers to the set $\mathcal{I}=${$u_i$} returned by the Meek separator algorithm (line 7 of Algorithm 1). For example, here in Figure 2, it returned $\mathcal{I}=${2}, meaning $u_i=2$. Evoking the termination condition of the algorithm (line 6 of Algorithm 1), this node $u_i=2$ corresponds to a Meek separator, i.e.,  {{2}} is a Meek separator.
> > >
> > > We understand that this notation might be a bit confusing with the previous usage of $\mathcal{I}$ as set of set. Therefore, we will revise all places in the paper that use calligraphical notations like $\mathcal{I}$ to set of set. This includes changing Figure 2 of the newly attached pdf to $\mathcal{I}$={{2}} and line 7 of Algorithm 1 to $\mathcal{I}=${{$u_i$}}.

---

> > > > ### Comment · Reviewer_BZ2Z · 2023-08-11
> > > > **Thank you for the response**
> > > >
> > > > I think all of my concerns are well-addressed. I will raise my score.

---

### Author Rebuttal · Authors · 2023-08-09

We thank all the reviewers for their insightful comments and suggestions.

---
In this general response, we attached a pdf of the additional figures that we will add to the manuscript. To summarize, this includes:

- A modified *Figure 1,* which now includes detailed explanations of our defined Meek separator and a separate panel illustrating the traditional graph separator. We will replace Figure 1 in the revised manuscript with this.
- *Figure 2,* which shows how Algorithm 1 finds the Meek separator in a 4-vertex DAG example. In this example, we walk through the iterations of Algorithm 1 while specifying realizations of which $u_i$ is picked in line 4 of Algorithm 1. We will add this example to section 4 of the revised manuscript.
- *Figure 3*, which illustrates a moral graph versus a graph that is not moral. We will add this example to Appendix B and reference to it when moral DAGs are introduced in section 2.
- *Figure 4*, which illustrates the essential graph, $\mathcal{I}$-essential graph, and connected components. We will add this example to Appendix B and reference to it when they are defined in section 2.

---

In addition, we’d like to provide further information regarding two common points raised by the reviewesr:

> **How the algorithm finds a Meek separator.**

In the *attached PDF*, we added a 4-vertex DAG example showing how Algorithm 1 finds the Meek separator in *Figure 2*. Further details can be found above and in the figure caption.

Intuitively, Algorithm 1 finds the Meek separator by doing *binary search* among the vertices of the inputted $1/2$-clique separator $K$. Lemma 10 establishes the existence of a subset of at most two vertices that form a Meek separator. It also shows that this subset must satisfy several nice properties. These properties empower us (as depicted in Algorithm 1) to identify such subsets using a binary search method within the vertices of the inputted 1/2-clique separator $K$.

To elaborate, since $K$ is a clique, its vertices have a natural order specified by the topological order of $K$. In each iteration, a vertex $u_i$ of $K$ is intervened, where the remaining vertices of $K$ are separated into $u_i$’s parents and children (i.e., before or after $u_i$ in the topological order). Then the algorithm moves on to search in the interval of the topological order ($K_i$ in Algorithm 1) that can potentially contain the Meek separator. This binary search procedure outputs a subset of vertices that satisfy the conditions of Lemma 10, which is guaranteed to form a Meek separator.

>**Statement, proof, and usefulness of Lemma 10.**

The statement of Lemma 10 is three-fold. Firstly, it states that there is a vertex u that fulfills the **constraint** “$|A_u| ≤ |V(\mathcal{G})|/2$ and $|A_v | > |V (\mathcal{G})|/2$ for all $v \in Des(u) \cap K$”. Secondly, it states that *any* vertex that fulfills this **constraint** will satisfy one of the two **conditions** “ 1). either $u$ is a sink vertex … 2) …”. Thirdly, it states that the $u$ (and potentially $x$) stated above corresponds to Meek separators.

Based on this statement, the proof proceeds in the following logic:

- Paragraph 1 (line 570-573) is showing that there exists a vertex $u$ that fulfills the **constraint**. Therefore the conditions asked by the Reviewer BZ2Z refer to “$|A_u| ≤ |V(\mathcal{G})|/2$ and $|A_v | > |V (\mathcal{G})|/2$ for all $v\in Des(u) \cap K$”. We will clarify this in the revised version by adding this description to line 573.
- Upon showing that the **constraint** can be fulfilled. We then try to show *any* vertex that fulfills this **constraint** will satisfy one of the two **conditions** “ 1). either $u$ is a sink vertex … 2) …”. Therefore paragraph 2 (line 574-575) starts by discussing if the $u$ that fulfills the constraint satisfies the first of the two conditions (i.e. if it is the sink node). Then paragraph 3 and onwards show that if $u$ does not satisfy condition 1, it will satisfy condition 2. We will clarify this in the revised version by adding these descriptions to line 574 and line 579.

The usefulness of this lemma lies in finding a vertex that satisfies the **constraint** (by using binary search, see response above regarding Algorithm 1). Subsequently, the **conditions** that such vertex satisfy lead to Meek separators. The division of two different conditions is for handling technicalities.


---
We provide the full list of references here:


[1] Karthikeyan Shanmugam, Murat Kocaoglu, Alexandros G. Dimakis, and Sriram Vishwanath. Learning Causal Graphs with Small Interventions. Advances in Neural Information Processing Systems, 28, 2015.

[2] Kristjan Greenewald, Dmitriy Katz, Karthikeyan Shanmugam, Sara Magliacane, Murat Kocaoglu, Enric Boix-Adserà, and Guy Bresler. Sample Efficient Active Learning of Causal Trees. *Advances in Neural Information Processing Systems*, 32, 2019.

[3] Jiaqi Zhang, Chandler Squires, and Caroline Uhler. Matching a desired causal state via shift interventions. *Advances in Neural Information Processing Systems*, 34:19923– 19934, 2021.

[4] Davin Choo and Kirankumar Shiragur. Subset verification and search algorithms for causal dags. *arXiv preprint arXiv:2301.03180*, 2023

[5] Yu, Kui, Lin Liu, and Jiuyong Li. "Discovering Markov blanket from multiple interventional datasets." *arXiv preprint arXiv:1801.08295* (2018).

---

### Decision · Program_Chairs · 2023-09-21

**Decision:**

Accept (poster)

**Comment:**

The authors propose Meek separators, basically intervention targets that reduce the remaining graph size, taking into account the Meek rules. Different from the most existing work, the authors' strategy is to identify this critical set of intervention targets with interventions. Compared to prior work, it does not provide an improvement on full graph discovery problem. But it nicely interpolates the existing solution by Choo 2022, to aiming to learn any specific subset of edges. It also has another application in the so-called causal (mean) matching problem that was proposed by Zhang 21.

The reviewers, although not very confident, unanimously recommended acceptance. Based on my own reading of the paper I believe there is sufficient merit and discoveries in the paper to warrant acceptance. One minor point is that the paper might be a bit hard to read for non-experts on this particular field, as noted by many reviewers. So it would be of great help if the authors can add the example in their rebuttal. I would actually instead recommend adding an example where skeleton is not complete, to better showcase the idea of Meek separator.

Thank you for the great work!